# Is In-Context Learning Sufficient for Instruction Following in LLMs?

**Hao Zhao**
EPFL

**Maksym Andriushchenko**
EPFL

**Francesco Croce**
EPFL

**Nicolas Flammarion**
EPFL

## Abstract

In-context learning (ICL) allows LLMs to learn from examples without changing their weights: this is a particularly promising capability for *long-context* LLMs that can potentially learn from *many* examples. Recently, Lin et al. (2024) proposed URIAL, a method using only three in-context examples to align base LLMs, achieving non-trivial instruction following performance. In this work, we show that, while effective, ICL alignment with URIAL still underperforms compared to instruction fine-tuning on the established benchmark MT-Bench, especially with more capable base LLMs. We then uncover the most relevant elements for successful in-context alignment, finding the crucial role of the decoding parameters. Based on these insights, we show that the approach of URIAL can indeed be improved by adding *high-quality*, possibly carefully selected via greedy search, demonstrations in context, getting closer to the performance of instruct models. Finally, we provide the first, to our knowledge, systematic comparison of ICL and instruction fine-tuning (IFT) for instruction following in the low data regime, where ICL can be a viable alternative to IFT. Overall, our work advances the understanding of ICL as an alignment technique and its relationship to IFT. We provide our code at https://github.com/tml-epfl/icl-alignment.

## 1 Introduction

The large-scale pre-training phase allows Large Language Models (LLMs) to acquire extensive knowledge and capabilities (Bubeck et al., 2023). However, these base models struggle to follow instructions directly from prompts, necessitating further fine-tuning to be used as conversational models. Traditional alignment methods include Supervised Fine-Tuning (SFT) on instruction datasets (IFT, Wei et al., 2021; Chung et al., 2022) or preference data (DPO, KTO, IPO, and ORPO, Rafailov et al., 2023; Ethayarajh et al., 2024; Azar et al., 2024; Hong et al., 2024), and reinforcement learning (RLHF and RLAIF, Ouyang et al., 2022; Bai et al., 2022). These approaches typically involve multiple rounds of refinement of the instructed model (Touvron et al., 2023), i.e. high computational cost, and need a large amount of annotated data like human preferences, which might be difficult to collect. However, a line of work (Taori et al., 2023; Chen et al., 2023; Zhou et al., 2023; Lee et al., 2023; Zhao et al., 2024; Kaur et al., 2024) has suggested that IFT on a small amount of high quality instruction-following examples, even only 1000, can be sufficient to match the performance of more complex alignment mechanisms. In particular, Zhou et al. (2023) introduced the Superficial Alignment Hypothesis, stating that LLMs acquire all their capabilities during pre-training, and fine-tuning only allows the models to better access such knowledge when interacting with users (Gudibande et al., 2023; Duan et al., 2023). Using such small instruction datasets for IFT has the clear advantage of significantly reducing the cost of model alignment.

Inspired by Brown et al. (2020) who showed that LLMs can learn from demonstrations provided as part of the input—a concept known as in-context learning (ICL)—Lin et al. (2024) have recently studied the feasibility of *in-context alignment* (Han, 2023; Li et al., 2023b). They found that including merely three carefully selected question-answer pairs in the prompt is sufficient to make *base* models follow instructions and interact with users at a similar level to instruction-fine-tuned models on their own benchmark. This fact strongly validates the idea that alignment can be achieved at low computational cost and with few examples, as well the Superficial Alignment Hypothesis, as it does not even require to modify the model weights. Moreover, in-context alignment is especially appealing since it opens up the possibility of customizing base models via ICL, i.e., without any fine-tuning,

which is potentially game-changing due to its simplicity and flexibility. While the results of Lin et al. (2024) are promising, they are restricted to a small number of base models and their own instruction following dataset.

**Analysis of in-context alignment.** In this work, we extend the evaluation of the URIAL (the abbreviation of Untuned LLMs with Restyled In-context ALignment) prompt strategy proposed by Lin et al. (2024) across several base models, including GPT-4-Base,[1] and on established instruction following benchmark MT-Bench (Zheng et al., 2023), see Table 1. First, we show that, although URIAL achieves reasonable performance, it still lags behind instruction-fine-tuned models. Then, to better understand the success but also weaknesses of in-context alignment, we analyze which are the most relevant ingredients in URIAL. We find that the decoding parameters in the LLMs generation (temperature, sampling scheme, etc.) may crucially influence the performance, and the optimal configuration allows even base models to achieve good scores on MT-Bench. Moreover, we show that randomly sampled triplets of examples from the highly curated instruction dataset SkillMix (Kaur et al., 2024) can achieve similar, or even better, results than URIAL when used for in-context alignment.

**Many-shot in-context alignment.** Then, we test various strategies to improve in-context alignment, leveraging recent

Table 1: **Systematic comparison of URIAL to aligned models on MT-Bench across different base LLMs.** For several recent model families, we compare the performance on instruction following tasks of the base LLMs plus URIAL (i.e. in-context alignment) to that of the instruct models, fine-tuned with sophisticated techniques like supervised instruction fine-tuning and RLHF. In most cases, the fine-tuned models outperform URIAL. * denotes the result taken from the URIAL GitHub repository.

| Model | 1st-turn | 2nd-turn | Average |
|---|---|---|---|
| Llama-2-7B + URIAL * | 5.75 | 3.91 | 4.83 |
| Llama-2-7B-Instruct | **7.14** | **5.91** | **6.53** |
| Llama-2-70B + URIAL * | **7.61** | 6.61 | 7.11 |
| Llama-2-70B-Instruct | 7.37 | **7.03** | **7.20** |
| Llama-3-8B + URIAL * | 6.84 | 4.65 | 5.75 |
| Llama-3-8B-Instruct | **8.29** | **7.42** | **7.86** |
| Llama-3-70B + URIAL * | 7.71 | 5.09 | 6.40 |
| Llama-3-70B-Instruct | **8.96** | **8.51** | **8.74** |
| Llama-3.1-8B + URIAL * | 6.95 | 5.31 | 6.13 |
| Llama-3.1-8B-Instruct | **8.27** | **7.73** | **8.00** |
| Mistral-7B-v0.1 + URIAL * | **7.49** | 5.86 | 6.67 |
| Mistral-7B-Instruct-v0.1 | 7.31 | **6.39** | **6.85** |
| Mistral-7B-v0.2 + URIAL * | 6.99 | 5.55 | 6.27 |
| Mistral-7B-Instruct-v0.2 | **8.06** | **7.21** | **7.64** |
| Mixtral-8x22B-v0.1-4bit + URIAL | 8.28 | 7.14 | 7.71 |
| Mixtral-8x22B-Instruct-v0.1-4bit | **8.78** | **8.25** | **8.52** |
| GPT-4-Base + URIAL | 7.96 | 5.04 | 6.50 |
| GPT-4 (March 2023) * | **8.96** | **9.03** | **8.99** |

models with *extensive context windows* which allow for longer in-context prompts. In particular, we study the effect of *many-shot* in-context learning by adding *high-quality* demonstrations from existing instruction datasets. Unlike what suggested by Lin et al. (2024), this approach can improve upon URIAL, but only when using high-quality examples, as those from SkillMix. However, it is still not sufficient to fully close the gap with aligned LLMs, as the performance plateaus after 10-30 in-context examples. This behavior is in contrast to many-shot ICL for tasks like summarization (Narayan et al., 2018), translation (Costa-jussà et al., 2022), or classification (Li et al., 2024), where providing many examples is clearly beneficial (Agarwal et al., 2024; Bertsch et al., 2024). We further test a simple greedy algorithm to select the in-context examples which optimize the MT-Bench score. This selection scheme outperforms, with 1 to 3 additional demonstrations, the many-shot approach with random samples, and allows to further reduce the distance from in-context aligned models to fine-tuned models.

**ICL vs IFT for alignment.** While our experiments suggest that ICL cannot match the instruction-following performance of models aligned through costly fine-tuning, possibly using preference data, it remains an open question whether ICL can compete with or outperform IFT in the low-data regime. We provide an extensive evaluation on multiple LLMs and datasets of both approaches when varying the question-answer dataset size between 3 and 4000 examples. We observe that, with high-quality data, ICL and IFT achieve almost identical 1st-turn MT-Bench score. Surprisingly, in the 2nd-turn score, IFT clearly outperforms ICL, with ICL performing even worse than the base model. This suggests that ICL overfits to the style of the examples shown in context.

---

[1] We received access to the base GPT-4 model via the OpenAI Researcher Access Program.

Overall, these results, complemented by several ablation studies, provide a more nuanced picture of ICL as an alignment technique compared to previous works. Moreover, our comparison of ICL to IFT when using the same data bridges a gap in the literature about understanding these orthogonal approaches for adapting pre-trained LLMs into conversational models. In summary, our contributions are as follows,

- **Analysis of In-Context Alignment** (Sec. 2): We systematically evaluate URIAL on a broader set of base LLMs, including GPT-4-Base, with established instruction-following benchmark. Our results indicate that in-context alignment with URIAL still underperforms instruction fine-tuning and more sophisticated alignment methods, and a proper decoding scheme is the crucial ingredient behind the empirical success of URIAL.

- **Scaling In-Context Alignment** (Sec. 3): We find that many-shot ICL with high-quality examples and a simple greedy algorithm can reduce the gap between in-context aligned models to fine-tuned models.

- **First systematic comparison of ICL vs IFT for instruction following** (Sec. 4): We show that ICL and IFT with the same number of examples are roughly equivalent for single-turn conversations in the low-data regime. However, IFT generalizes substantially better than ICL when more examples are present, especially for multi-turn conversations.

## 2 UNCOVERING THE LIMITS OF ICL FOR INSTRUCTION FOLLOWING

In the following, we provide an in-depth analysis which aims at (1) systematically comparing the performance of base models plus URIAL to that of aligned models (Sec. 2.1), (2) understanding which components of URIAL (and in-context alignment in general) are most important for its effectiveness (Sec. 2.2), (3) testing the influence of question-answers format in our task (Sec. 2.3).

### 2.1 SYSTEMATIC EVALUATION OF URIAL

**Background on URIAL.** Lin et al. (2024) propose to use a prompt with as few as three constant stylistic examples, which follow a curated answering structure that human users are familiar with, and a carefully designed set of rules, highlighting the format and principles AI assistant needs to follow strictly when generating answers, to align base LLMs with ICL. We present the complete URIAL prompt in Fig. 9. The highly curated examples always begin with affirming the user query by saying something like "Hi, I'm happy to help you.", then proceed to answer the query by enumerating necessary items and steps, and conclude with an engaging summary. The set of rules first elucidate the scenario and format of the subsequent interaction between the human user and the AI assistant, then outline multiple qualitative aspects, including helpfulness, honesty, and harmlessness, which the base LLMs should adhere to when answering queries.

**Experimental setup.** We compare the performance of several base models with the URIAL in-context prompt to that of their instruction fine-tuned versions. Table 1 shows the results on MT-Bench (Zheng et al., 2023), one of the most popular benchmarks for instruction following ability of LLMs. We note that Lin et al. (2024) originally tested URIAL on their own dataset, in which only $8\%$ of the examples are obtained from MT-Bench. We compare several LLMs of different sizes and capabilities, ranging from Llama-2-7B (Touvron et al., 2023) to the *base* GPT-4 model (OpenAI, 2023). For simplicity, we consider the *official* instruction-tuned LLMs, i.e., provided from the same source of the base models, even if there exist third-party fine-tuned versions which can achieve higher scores.

**Results.** In Table 1 we observe that base models with URIAL achieve competitive performance on the 1st-turn score, but cannot match that of their instruction fine-tuned counterparts in all cases except for Llama-2-70B and Mistral-7B-v0.1 (both originally included in Lin et al. (2024), unlike most others). Conversely, the 2nd-turn score with URIAL is *significantly worse* than for instruction-tuned LLMs: we hypothesize that this is because no multi-turn demonstrations are given in context. Indeed, Zhou et al. (2023) show that, even in IFT, having a few multi-turn training examples significantly improve the performance of the model on multi-turn conversations. Because of this fact, in the rest of the paper, we mainly focus on the 1st-turn score and single-round conversations, and track of 2nd-turn performance for completeness. To fill in the gap in the influence of multi-turn conversations, we study multi-turn alignment via ICL in Sec. 2.4. Finally, we detail the breakdown of the results over categories in Table 11 (see App. E.4 for details).

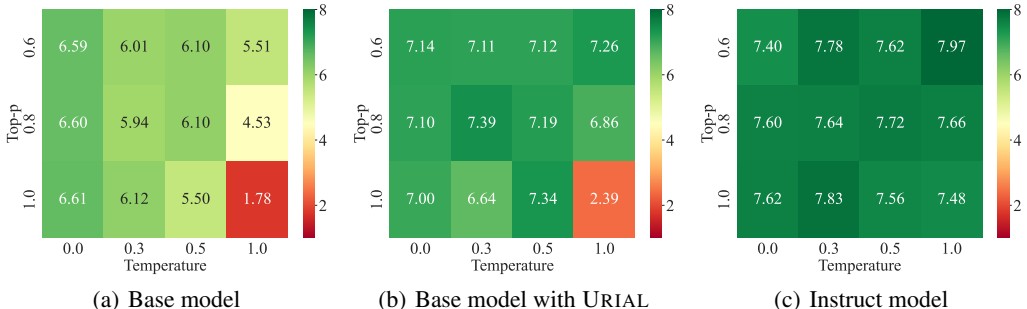

(a) Base model     (b) Base model with URIAL     (c) Instruct model

Figure 1: **Effect of decoding parameters on the 1st-turn MT-Bench scores across models.** We vary temperature and top-$p$, with a fixed repetition penalty of 1.15 as used in the URIAL codebase. The heatmaps show that the answering quality of the base model (Mistral-7B-v0.2) with and without URIAL is sensitive to the decoding schemes. Conversely, the performance of the instruct model is robust to varying the decoding parameters. Surprisingly, with proper decoding parameters, the base model alone is already capable of following instructions. See the complete results in App. E.3.

## 2.2 WHAT MATTERS FOR IN-CONTEXT ALIGNMENT VIA URIAL?

**Decoding parameters.** When computing the performance of URIAL, Lin et al. (2024) use decoding parameters (temperature = 0, top-$p$ = 1, repetition penalty = 1.15) which differ from the default ones in MT-Bench (where temperature depends on the topic of the question, top-$p$ = 1, repetition penalty = 1). To clarify the influence of these parameters that were not explicitly discussed in Lin et al. (2024), we compute the performance of base LLM, base LLM with URIAL, and fine-tuned LLM when varying decoding configurations. In Fig. 1 we fix the repetition penalty to 1.15, and vary temperature and top-$p$, for the Mistral-7B-v0.2 models. Surprisingly, we notice that with the decoding parameters of URIAL (temperature = 0, top-$p$ = 1) even the base model without any in-context example achieves reasonable MT-Bench score (6.61 vs 7.00 of URIAL). Moreover, the temperature value appears to have the most influence of the performance of the base model. With URIAL, almost all configurations provide very similar results, with the exception of temperature = 1, top-$p$ = 1 (possibly because it allows the sampling of generated tokens from the tail of the token distribution, thus producing low-quality text). Finally, the aligned model performs similarly across all decoding schemes, with slightly better results than URIAL. These results suggest that (1) the decoding parameters are an *overlooked factor* contributing to the success of in-context alignment, and (2) fine-tuning adjusts the sampling distribution of language models so that even with high-variance decoding configurations the generated text preserves high quality. We provide additional analysis for no repetition penalty (i.e. = 1) in Fig. 13 and Llama-3.1-8B in Fig. 14 in App. E.3: these extensive experiments further validate, with different base LLMs and settings, that the decoding scheme is a crucial factor for the instruction-following behavior of base models, and also affects, but to lower degree, in-context alignment. Instead, the fine-tuned models are robust to variations in the decoding parameters. Then, for the rest of the experiments we fix the decoding scheme of URIAL.

**In-context prompt.** Next, we want to disentangle the influence of the various elements of URIAL on instruction following performance. In Fig. 2 we track the MT-Bench scores when putting in-context all possible subsets of the three demonstrations of URIAL (i.e. from 0 to 3 examples are used). Moreover, for each case we test either adding or not the part of the prompt including the rules to follow. First, we find that neither 1st-turn nor 2nd-turn MT-Bench scores are significantly influenced by the addition of the set of rules, in turn highlighting the importance of the examples. Focusing on 1st-turn score (left plot in Fig. 2), we see that increasing the number of in-context demonstration progressively improves the performance of the base model (Mistral-7B-v0.2 in this case): this observation motivates us to add further high-quality in-context demonstrations, see Sec. 3. Finally, the 2nd-turn has the opposite trend, getting degraded by more (single-round) demonstrations.

## 2.3 IMPORTANCE OF QUESTION-ANSWER MATCHING

Surprisingly, Min et al. (2022) showed that using demonstrations with *random labels* does not significantly impair the results of ICL on classification and multiple choice tasks. We conduct a similar study for instruction following, see Table 2. Denoting $\{(X_i, Y_i)\}_i$ the set of in-context examples,

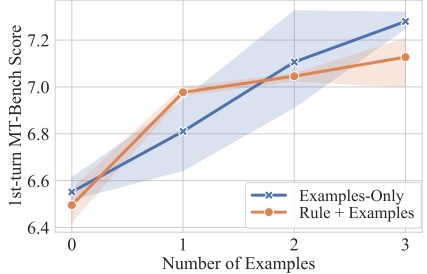 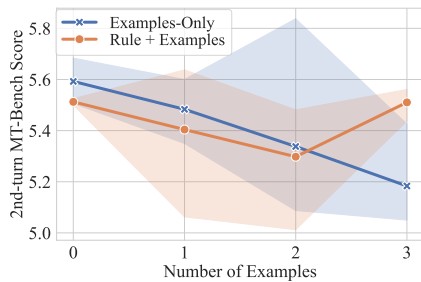

Figure 2: **Influence of the individual components of URIAL.** We report 1st and 2nd turn MT-Bench score for every subset of the three demonstrations of URIAL with Mistral-7B-v0.2. We test each configuration with ("Rule + Examples") and without ("Examples-Only") the URIAL set of rules in the in-context prompt. We observe a clear increasing trend in 1st-turn score with more examples, but a decrease in the 2nd-turn performance. The set of rules does not seem to influence the results. The randomness for 0 and 3 examples is caused by the small fluctuations in the score of the GPT-4 judge.

Table 2: **Importance of question-answer matching of demonstrations for in-context alignment.** We report the 1st-turn MT-Bench score of Mistral-7B-v0.2 and Llama-3.1-8B when varying the structure of the in-context examples $\{(X_i, Y_i)\}_i$, where $X_i$ is the query and $Y_i$ is the corresponding ground-truth answer.

| In-context prompt | URIAL (3 examples) | | URIAL + Greedy Search (6 examples) | |
|---|---|---|---|---|
| | Mistral-7B-v0.2 | Llama-3.1-8B | Mistral-7B-v0.2 | Llama-3.1-8B |
| no demonstrations | 6.52 | 6.25 | 6.52 | 6.25 |
| $X$ only (no $Y$) | 5.90 | 4.48 | 5.53 | 5.10 |
| $Y$ only (no $X$) | 6.94 | 6.83 | 7.02 | 6.98 |
| circular shift of $Y$ | 5.04 | 5.69 | 5.59 | 2.13 |
| in-domain random $Y$ | 6.24 | 6.26 | 4.63 | 5.49 |
| out-of-domain random $Y$ | 4.13 | 3.73 | 1.50 | 4.36 |
| correct $Y$ | **6.99** | **6.95** | **7.43** | **7.81** |

with $X_i$ the query and $Y_i$ the corresponding ground-truth answer, we test several configurations on two sets of $\{(X_i, Y_i)\}_i$: the 3 URIAL examples and the 3 URIAL examples with the 3 examples found by our greedy search from Sec. 3.2 (to check the effect of increasing the number of demonstrations).

First, we do not use any demonstration, i.e., the original base models: with the decoding parameters found in the previous section, this already achieves reasonable results. Surprisingly, providing *the questions without the answers* (second row in Table 2) degrades the performance, while the opposite, *answers only*, is effective (0.4-0.7 higher scores than the base model). Next, we permute the answers $Y_i$s with a circular shift of one position, so that all correct answers are still contained in the prompt but matched with the wrong question: this significantly degrades the performance, especially with more examples (URIAL + Greedy Search), e.g. Llama-3.1-8B attains a score of only 2.13 (note that the minimum score is 1). Also, for each question, we sample a new answer from those provided for other instructions in the same category (*in-domain*): e.g., a question about coding is paired with an answer from a different coding question, ensuring that, while the content may be incorrect, the style remains appropriate. Although worse than the original one, this configuration achieves decent scores. Finally, we sample answers from instructions belonging to different *out-of-domain* categories, so that even their style does not match what expected for each question: this leads to the worst performance.

These results show that not all the conclusions from Min et al. (2022) apply to ICL used for instruction following. Most importantly, using answers with correct content and, especially, correct style is crucial for the success of ICL. This property becomes even more evident when increasing the number of examples (to 6 instead of 3), and motivates us to use a highly-curated instruction dataset like SkillMix in experiments in the next sections. Finally, this result suggests that ICL for instruction following works differently compared to other less open-ended tasks.

## 2.4 MULTI-TURN ALIGNMENT VIA ICL

In IFT, Zhou et al. (2023) find that adding a few multi-turn examples to the training data significantly improves the model's performance on multi-turn conversations. URIAL (Lin et al., 2024) shows that

Table 3: **In-context alignment with *multi-turn* examples.** Second-turn answers for all methods are generated using GPT-4o. Extending URIAL with two-turn examples improves the 2nd-turn MT-Bench score. This shows that in-context alignment is not limited to single-turn performance.

| In-context prompt | Mistral-7B-v0.2 | | | Llama-3.1-8B | | |
|---|---|---|---|---|---|---|
| | Turn 1 | Turn 2 | Avg | Turn 1 | Turn 2 | Avg |
| Base model (no prompt) | 6.52 | 5.68 | 6.10 | 6.25 | 5.17 | 5.71 |
| Original URIAL (single-turn) | 6.99 | 5.55 | 6.27 | 6.95 | 5.31 | 6.13 |
| Two-turn URIAL | 7.18 | 5.68 | 6.43 | **7.31** | 6.44 | 6.87 |
| URIAL + 3 two-turn SkillMix examples (no separation tag) | **7.46** | 5.88 | 6.67 | 7.21 | 6.43 | 6.82 |
| URIAL + 3 two-turn SkillMix examples (separation tag) | 7.20 | **6.31** | **6.76** | 7.22 | **6.71** | **6.97** |
| URIAL + 6 two-turn SkillMix examples (no separation tag) | 7.18 | 5.89 | 6.53 | 7.14 | 6.26 | 6.70 |
| URIAL + 6 two-turn SkillMix examples (separation tag) | 7.24 | 5.78 | 6.51 | 6.89 | 6.33 | 6.61 |

the base LLMs can handle multi-turn conversations even with only having single-turn demonstrations in the context, but it does not quantify the performance nor rigorously study the role of examples that have more than one turn of conversation. To bridge this research gap, we further test how the presence of multi-turn examples in the context influences the instruction-following performance of in-context aligned models.

**Experimental setup.** We experiment using two open source LLMs, Mistral-7B-v0.2 and Llama-3.1-8B, and evaluate on MT-Bench. To ensure the multi-turn examples we test have high quality, we select single-turn examples randomly from SkillMix as the basis, the first turn, and then prompt GPT-4o to ask a follow-up question after seeing the first-turn conversation and generate an answer by using an instruction "Here is a single-turn conversation between a user and an assistant, please ask a follow-up question and provide an answer.". We examine the influence of two-turn examples from two perspectives, the number of two-turn examples (3 vs 6), and the format, with or without separation tags like "Here are 3 single-turn/multi-turn examples:".

**Results.** In Table 3, we observe that extending 3 URIAL single-turn examples to 2-turn examples consistently improves the evaluation scores on both turns, showing that in-context alignment solely with multi-turn examples can take care of single-turn conversations. The presence of two-turn examples in the context always leads to improved 2nd-turn MT-Bench scores, for example, an increase from 5.31 to 6.43 after adding 3 two-turn SkillMix examples on top of URIAL on Llama-3.1-8B. Another finding from our experiments is that the format of placing demonstrations plays a pivotal role in multi-turn interactions. In particular, adding a separation tag to arrange demonstrations in two groups improves 2nd-turn scores, from 5.88 to 6.31 on Mistral-7B-v0.2 and from 6.43 to 6.71 on Llama-3.1-8B, although sometimes it hurts the 1st-turn performance. However, we find that increasing the number of two-turn examples in the context does not lead to more benefits with or without a better format.

## 3 MANY-SHOT IN-CONTEXT LEARNING FOR INSTRUCTION FOLLOWING

Sec. 2.1 indicate that URIAL alone is not able, in most cases, to reach the performance of instruct models. In the following, we explore whether we can close this gap. We focus on Mistral-7B-v0.2 (Jiang et al., 2023) and Llama-3.1-8B (Dubey et al., 2024) since (a) both URIAL and the aligned model achieve competitive performance in Table 1, and (b) these base models have context windows of 32k and 128k respectively, which can fit many examples (around 50 for Mistral-7B-v0.2 and more than 200 for Llama-3.1-8B). Details on the experimental setup are in App. A.

**Data.** We adopt the SkillMix (Kaur et al., 2024) dataset consisting 4,000 examples as the source of high-quality examples for our many-shot in-context alignment experiments. The query-answer paired data generation process of SkillMix explicitly combines diversity and difficulty through random skill combinations, where the set of core skills for instruction-following, such as developing training programs and digital proficiency, are extracted by prompting a powerful LLM like GPT-4-Turbo. We present more details of SkillMix, including some examples (Fig. 6 and Fig. 7), in App. A.1.

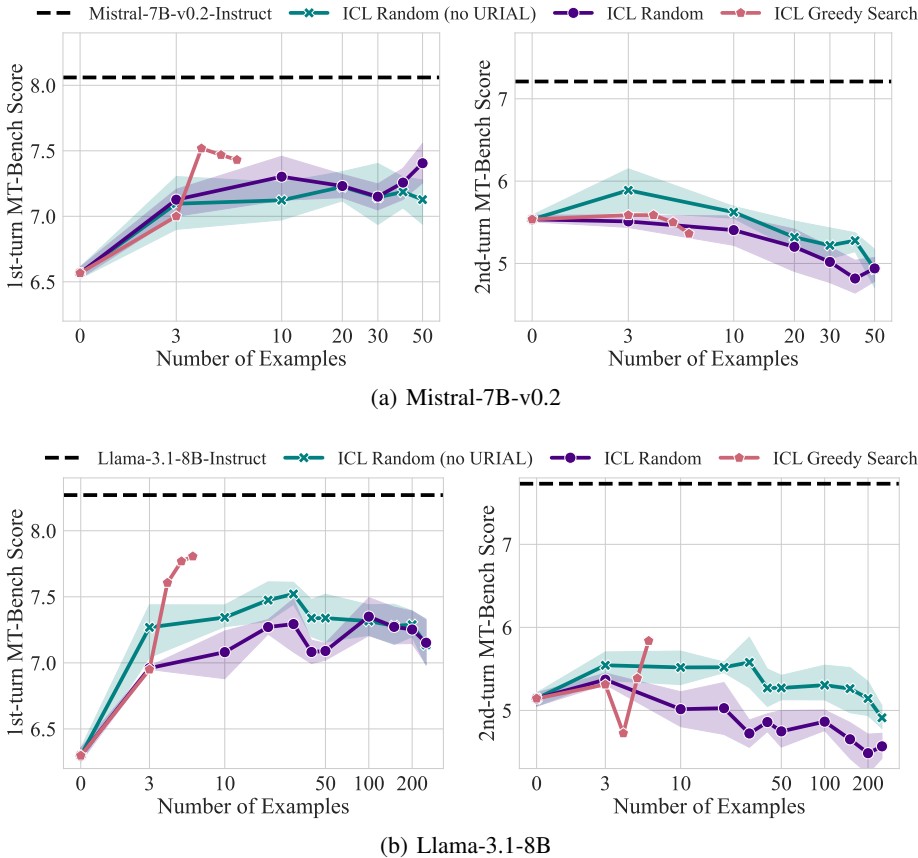

Figure 3: **Scaling the number of demonstrations for alignment with ICL on Mistral-7B-v0.2 and Llama-3.1-8B.** We measure the alignment performance of different settings using the MT-Bench score. ICL with more demonstrations quickly saturates and does not bridge the performance gap between the base model and its aligned counterpart. In particular, the ICL alignment performance of 3 random examples from the high-quality SkillMix dataset surpasses that of 3 examples from URIAL.

### 3.1 SCALING UP THE NUMBER OF IN-CONTEXT DEMONSTRATIONS

Given the success of many-shot ICL (Agarwal et al., 2024), we test the effect of increasing the number of in-context demonstrations. We sample random examples from the SkillMix dataset, since it contains high-quality question-answer pairs. We consider two scenarios: (1) all demonstrations are randomly chosen from SkillMix (the set of rules from URIAL is kept), and (2) we add the new demonstrations on top of the URIAL examples. In Fig. 3, we report the results on MT-Bench when varying the number of demonstrations. We repeat sampling with 5 random seeds and show mean and standard deviation over the corresponding results. We observe that for both base LLMs, the 1st-turn score increases with 3, 10 and 20 examples, but then plateaus without clear benefits from scaling up beyond 30 demonstrations (for Llama-3.1-8B the trend becomes even slightly negative with more than 100 examples). Thus, we find that adding up to 30 high-quality examples from SkillMix can improve alignment via ICL. This result contrasts with the findings of Lin et al. (2024), which showed that URIAL performed better with 3 demonstrations than with 8.

Next, we observe that in-context alignment does not improve the second-turn score. In fact, adding more examples continues to decrease performance, even falling below that of the base models. This behavior is likely due to the presence of only single-turn examples in the prompt, causing the LLM to respond in the same style without adapting to more complex conversations. Overall, these results show that, unlike in the setting of Agarwal et al. (2024); Bertsch et al. (2024), simply scaling the number of ICL examples is not sufficient to consistently improve the instruction following performance.

Finally, we notice that on Llama-3.1-8B, using three examples from SkillMix attain, on average, better performance than using the three examples of URIAL. Overall, there is no significant difference

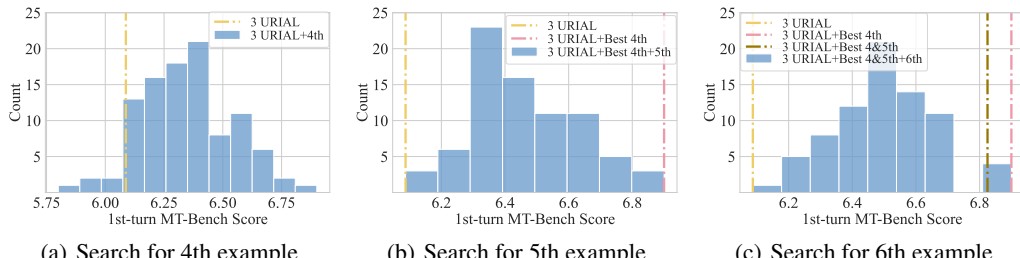

(a) Search for 4th example  (b) Search for 5th example  (c) Search for 6th example

Figure 4: **The distribution of the 1st-turn MT-Bench score (GPT-4-Turbo as judge) on Llama-3.1-8B obtained by adding multiple instructions from SkillMix (Kaur et al., 2024) as a 4th (a), 5th (b), 6th (c) demonstration to URIAL.** The dashed lines of various colors refer to the 1st-turn MT-Bench score of the obtained searching results. A majority of the 4th examples contribute positively to the model's instruction-following performance, but the improvement quickly diminishes when running the greedy search for 5th and 6th demonstrations.

Table 4: **Improved in-context (IC) alignment by adding to URIAL prompt the demonstrations found via greedy search.** We find the optimal IC prompts in an incremental way by selecting query-response pairs from the high-quality SkillMix dataset. We report the 1st-turn score on MT-Bench and the length-controlled (LC) win rates on AlpacaEval 2.0 when using different IC prompts.

| Model | Mistral-7B-v0.2 | | Llama-3.1-8B | |
|---|---|---|---|---|
| | MT-Bench (1st) | AlpacaEval 2.0 | MT-Bench (1st) | AlpacaEval 2.0 |
| URIAL (3 examples) | 7.00 | 8.22 | 6.95 | 7.28 |
| URIAL + greedy search (1 ex.) | **7.52** | 7.53 | 7.61 | **8.61** |
| URIAL + greedy search (2 ex.) | 7.47 | 7.78 | 7.77 | 8.16 |
| URIAL + greedy search (3 ex.) | 7.43 | **8.55** | **7.81** | 8.19 |

in the scaling behavior between using or not using the URIAL demonstrations. Thus, we conclude that the success of in-context alignment is only partially dependent on the demonstrations themselves, provided they are of sufficient quality (see also Sec. 4). We also include the results of scaling experiments (Sec. E.1) conducted on AlpacaEval 2.0, which consists of 805 examples.

## 3.2 GREEDY SEARCH FOR EFFECTIVE DEMONSTRATIONS

Given the limited success of adding random demonstrations to URIAL, we try to greedily maximize the MT-Bench score by testing 100 high quality instructions from SkillMix as the 4th additional example to URIAL. For each resulting ICL prompt, we compute the MT-Bench score with GPT-4-Turbo as judge instead of GPT-4 (used in MT-Bench) as the former is faster, cheaper, and helps mitigate potential overfitting to the benchmark score. We then repeat this procedure sequentially to find a 5th and a 6th demonstration, restricting the search space at each round to only instructions leading to a high enough MT-Bench score to reduce the computational cost (see details in App. A.3).

We add the results of the greedy search to the plots in Fig. 3, computing the true MT-Bench (i.e., with GPT-4 as judge). For both base models, the 4th example found by greedy search is sufficient to match the best 1st-turn score achieved by scaling the in-context examples with random demonstrations (see Sec. 3.1). For Llama-3.1-8B, the 5th and 6th demonstrations further improve the score, while they are not helpful for Mistral-7B-v0.2. In Table 4, we provide the details of these results: the 4th ICL example yields a significant improvement over URIAL, e.g., from 6.95 to 7.61 for Llama-3.1-8B, with only a further +0.20 given by the 5th and 6th. Overall, this evaluation further indicates that the improvement in instruction following one can achieve with in-context alignment quickly saturates when increasing the number of examples. In Fig. 4, we show the distribution of the scores (with GPT-4-Turbo as judge) at step of the greedy search when using Llama-3.1-8B as base model (the analog for Mistral-7B-v0.2 in App. E.2). Most demonstrations improve the score when added as the 4th example on top of URIAL (as shown in the first plot), but no further progress is observed with additional demonstrations.

We further test the IC prompts found by the greedy search on the AlpacaEval 2.0 (Li et al., 2023a) benchmark, where we measure length-controlled win rate. As shown in Table 4, using the examples

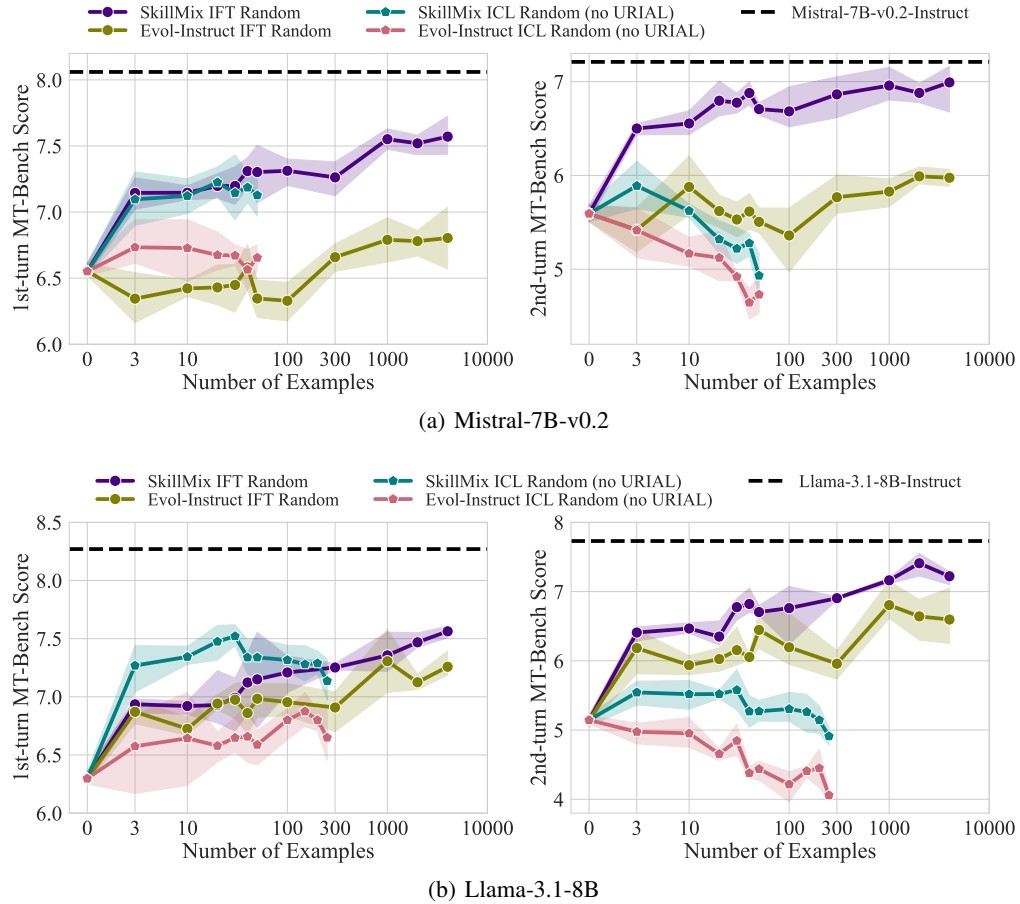

Figure 5: **Comparison of ICL vs IFT for alignment in the low data regime.** We measure the alignment performance of different settings for Mistral-7B-v0.2 and Llama-3.1-8B using the MT-Bench score. IFT with more demonstrations keeps improving the alignment performance, almost bridging the gap between the base model and its aligned counterpart. IFT-aligned models perform well on multi-turn conversations, unlike with ICL. Finally, data quality has significant impact on both IFT and ICL: the higher-quality SkillMix leads to better performance than Evol-Instruct.

given by greedy search leads to better results than plain URIAL on AlpacaEval 2.0, without directly optimizing for it. This improvement indicates that the found IC prompt does not completely overfit to the MT-Bench score, although the gains are less consistent than on MT-Bench. For Mistral-7B-v0.2, URIAL with 3 greedy search examples is the only configuration that outperforms URIAL, while for Llama-3.1-8B just one additional demonstration yields the best win rate, with a significant improvement over URIAL (7.28 vs 8.61).

## 4 A COMPARISON OF ICL VS IFT FOR INSTRUCTION FOLLOWING

The results from the previous sections strongly suggest that in-context alignment might not be as competitive as sophisticated, computationally heavy alignment techniques. However, it is unclear how it compares to more lightweight approaches, such as IFT, when applied to a small number of examples. In the low-data regime, the choice between ICL and IFT can be viewed as a decision between allocating resources for fine-tuning (which permanently modifies the model's weights) or increasing inference time by adding the in-context prompt each time.

**Setup.** We test IFT on the same number of instructions from SkillMix as used in the many-shot ICL experiments in Sec. 3, ranging from 3 to 250 (the maximum allowed by the context length of the base models). Moreover, we scale the dataset up to 4k, that is the entire SkillMix. Each training run is repeated over multiple seeds, resulting in different training datasets (more details in App. A.4).

**Results on Mistral-7B-v0.2.** The top part of Fig. 5 shows that IFT and ICL on SkillMix (violet and green curves respectively) perform almost identically in 1st-turn score until 50 examples are used. Beyond this point, increasing the number of training examples consistently improves the performance of IFT. The strong performance of IFT in the low-data regime is particularly surprising, as one might expect overfitting when training on very few examples (as few as 3) over several epochs, yet this does not occur. Finally, ICL and IFT show nearly opposite trends for the 2nd-turn score: increasing the dataset size benefits IFT (almost reaching the performance of the instruct model) but detrimental to ICL. This behavior suggests that, while IFT is less flexible due to the model weights being updated, it generalizes better to tasks different from those used for alignment (recall that the instructions in SkillMix are single-turn only).

**Results on Llama-3.1-7B.** As shown in the bottom part of Fig. 5, ICL outperforms IFT when using between 3 and 30 demonstrations, though IFT remains effective even with these few examples on this base model. However, we note that IFT matches or exceeds the best performance of ICL (obtained with 20-30 examples) only when using 2k or 4k examples (the entire SkillMix), which represents two orders of magnitude more data. This result confirms that ICL with high-quality data is a viable alternative to IFT when only a limited number of demonstrations are available. Finally, the observations for the 2nd-turn scores are consistent with those for Mistral-7B-v0.2, with IFT consistently outperforming ICL.

**Effect of data quality.** Finally, we test the effect of using lower-quality instructions compared to SkillMix. In this experiment, we repeat the ICL vs. IFT comparison using random demonstrations from Evol-Instruct-70k (Xu et al., 2024), and show the results in Fig. 5. With Evol-Instruct data, both ICL (red curve) and IFT (yellow curve) perform significantly worse than their counterparts using SkillMix. This trend is consistent across the number of examples, base models, and for both single-turn and multi-turn instructions. Interestingly, the performance gap between IFT on SkillMix and Evol-Instruct is significantly smaller on Llama-3.1-8B than on Mistral-7B-v0.2, suggesting that better pre-trained models may partially compensate for lower-quality fine-tuning data. Finally, we notice that ICL with 3 examples from Evol-Instruct gets worse 1st-turn scores than URIAL (with also 3 examples), whereas ICL with 3 examples from SkillMix matches (on Mistral-7B-v0.2) or outperforms (on Llama-3.1-8B) URIAL (see also discussion in Sec. 3.1). These observations further confirm the importance of instruction quality for alignment, whether in the context of in-context learning or instruction fine-tuning.

## 5 CONCLUSIONS

In this work, we have first illustrated that ICL via URIAL is a good baseline for instruction-following alignment, but with a few limitations: it typically performs slightly worse than IFT on single-turn conversations and does not generalize well to multi-round ones. Then, we have uncovered the key components for IC alignment: e.g. the decoding parameters alone can, surprisingly, make base models reasonably good at instruction following. Adding in-context high-quality demonstrations improves performance beyond what previously suggested (Lin et al., 2024), but not enough to reach LLMs aligned with sophisticated methods (e.g., RLHF). We conjecture that via ICL, the LLM can learn to infer the response style, but the overall learning signal is not sufficiently strong to benefit from a large amount of examples, despite the long context windows of recent LLMs.

Moreover, to the best of our knowledge, we have provided the first systematic comparison of ICL and IFT for instruction following when using the same (small) number of demonstrations. Surprisingly, the two approaches are roughly equivalent in terms of single-turn conversation performance. However, models trained via IFT, unlike ICL, can generalize to multi-round conversations, which are out-of-distribution compared to the examples used for alignment. Overall, our work provides a deeper and more complete understanding of how in-context alignment works, as well as of its potential and limitations, in particular in comparison to fine-tuning. This comprehension might be the basis for future work aimed at using ICL to efficiently customize LLMs without the need of fine-tuning, as well as exploring the fundamental differences between base and aligned models.

**Limitations.** We perform greedy search with limited computational resources, e.g. the set of candidate examples is relatively small. We expect that more effective ICL prompts could be identified with additional resources. Moreover, our focus is on instruction-following tasks, but other types of alignment, such as safety-oriented tasks, may also be of interest.

## ACKNOWLEDGEMENTS

We are grateful to OpenAI for providing us API credits and access to GPT-4-Base as part of the OpenAI Researcher Access Program. M.A. was supported by the Google Fellowship and Open Phil AI Fellowship. This work was partially funded by an unrestricted gift from Google and by the Swiss National Science Foundation (grant number 212111).

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

## A    EXPERIMENTAL DETAILS

In this section, we show more details about the experiments we conducted in the paper.

### A.1    DATASETS AND BASE MODELS

We select instruction-following demonstrations for the base LLMs to learn in context from open-sourced instruction fine-tuning datasets: (a) SkillMix-4k (Kaur et al., 2024) is an automated approach for synthesizing diverse, high-quality IFT data. It primarily involves two stages: first leveraging the LLMs to propose a set of critical "skills" for instruction-following, from which a pair of skills are randomly chosen to facilitate synthetic data generation based on powerful LLMs. We show two examples from SkillMix in Fig. 6 and Fig. 7. (b) Evol-Instruct-70k (Xu et al., 2024) contains 70k training examples with varying complexity and is well-known for the use to build the series of WizardLM models. SkillMix-4k is the primary dataset for all experiments in the paper and Evol-Instruct-70k is solely used for scaling experiments and ablation study of data quality.

We compare ICL performance across multiple models that have sufficiently large context windows:

1. Mistral-7B-v0.2 (Jiang et al., 2023) has 7.3 billion model parameters. On many established benchmarks, it outperforms LLMs that have significantly more parameters, such as Llama-2-13B and Llama-1-34B. The trained context length of Mistral-7B-v0.2 is 32k tokens.
2. Llama-3.1-8B (Dubey et al., 2024) is the newest and most powerful 8B model from Meta when we are writing, and it supports multilingual dialogue use cases. It supports input texts containing up to 128k tokens.

In some experiments, the following base LLMs are used:

1. Llama-2-7B-80k (Fu et al., 2024) is a variant of Llama-2-7B model fine-tuned with 80k context on a carefully designed long-document data mixture.
2. Mixtral-8x22B-v0.1-4bit is a variant of Mixtral-8x22B-v0.1 (Jiang et al., 2024), which is a pre-trained generative sparse Mixture of Experts, quantized with 4-bit precision. It contains $\sim$176B parameters and $\sim$44B active during inference, and it has a 65k context window.

We use the same decoding configuration as what is used in URIAL (Lin et al., 2024). Concretely, we employ greedy decoding, i.e., temperature $= 0.0$, for all models, including base and instruction fine-tuned models, to maximize reproducibility and secure a fair and robust evaluation. Top-p $= 1.0$ is adopted to keep the full cumulative probability distribution. Besides, we use repetition penalty $= 1.15$[2] on base models to prevent degeneration.

### A.2    EVALUATION

**MT-Bench** (Zheng et al., 2023), consists of 80 high-quality and challenging questions that have two-round interaction, designed to examine the multi-turn conversation and instruction-following capability of models. It features 8 common categories of user prompts: coding, math, reasoning, extraction, roleplay, writing, humanities/social science, and STEM.

**AlpacaEval 2.0** (Li et al., 2023a) provides 805 test instructions, on which we generate new responses using the target model, and then calculate the score by competing with the baseline model (i.e., GPT-4-Turbo) judged by a designated automatic evaluator. We apply the AlpacaEval 2.0 benchmark in our experiments to ensure that the effective demonstrations found through greedy search don't overfit to MT-Bench.

### A.3    GREEDY SEARCH

Firstly, we randomly sample 100 examples from the high-quality IFT dataset, SkillMix-4k, and then create 100 new 4-shot prompts by adding each one of the 100 sampled examples, as the fourth

---

[2]As in the codebase at https://github.com/Re-Align/URIAL/blob/main/run_scripts/mt-bench/_run_mt_bench.sh.

### Skills: developing training programs, digital proficiency

### Instruction
I'm planning to develop a digital literacy program specifically for seniors at our community center. Could you guide me on the essential components to include, how I can leverage digital tools effectively, and provide some concrete examples or success strategies from similar programs?

### Response
Creating a tailored digital literacy training program for seniors is key to enhancing their technology engagement. Below is a comprehensive guide featuring improved content based on earlier feedback:

**1. Assessing Learning Needs:**
Conduct surveys or focus groups to understand seniors' technological backgrounds and learning objectives. This data will guide the customization of training content.
**2. Structuring the Program:**
Organize the program into manageable modules:
- **Basic Computing Skills:** Instructions on fundamental operations of devices, incorporating user-friendly technology like Chromebooks, which are simpler for beginners.
- **Internet Navigation:** Teach safe browsing, using straightforward and secure websites for everyday tasks.
- **Email and Communication Tools:** Demonstrate setting up and using email and video calling, with adjustments for larger display settings to aid visibility.
- **Online Safety:** Educate on creating strong passwords with tools like LastPass, identifying secure websites, and recognizing scams.
**3. Using Digital Tools in Training:**
Integrate specific digital tools to enhance learning:
- **Interactive Tutorials:** Implement platforms like GCFGlobal that offer senior-friendly interactive lessons.
- **Feedback Mechanisms:** Use simple, engaging quizzes through Kahoot! to review and reinforce learning.
**4. Scheduling and Accessibility:**
Plan short, regular sessions. Ensure the learning environment is comfortable and accessible, considering both physical and online settings.
**5. Facilitating Engagement:**
- **Peer Learning:** Support seniors working in pairs to promote mutual assistance and engagement.
- **Ongoing Support:** Establish a support system such as a helpline or dedicated email for ongoing help, coupled with monthly follow-up sessions.
**6. Case Studies and Success Stories:**
Refer to successful programs like AARP's TEK workshops, highlighting their positive impact on senior confidence with technology.
**7. Measure Impact and Adapt:**
Use analytics tools to monitor participation and feedback, continuously refining the program based on empirical data.

By focusing on these enhanced, specific steps and maintaining a dynamic feedback loop, your digital literacy program will significantly improve seniors' technological fluency and confidence.

### Query Type: Planning

### Data Generator: gpt-4-turbo-2024-04-09

Figure 6: An example of SkillMix describes a conversation about planning and includes two skills, developing training programs and digital proficiency.

### Skills: performance metrics analysis, writing review articles

### Instruction
I am working on a review article about various performance metrics used in machine learning models for a class project. Specifically, I want to understand the real-world implications and methodological differences between the F1 Score and AUC-ROC. Can you explain and compare these metrics based on the following excerpt from a research paper?

Excerpt:
"In evaluating classifier performance, the F1 Score and AUC-ROC are pivotal metrics. The F1 Score, a harmonic mean of precision and recall, crucially balances the trade-offs between false positives and false negatives. On the other hand, AUC-ROC, which measures the area under the receiver operating characteristic curve, provides an aggregate measure of performance across all classification thresholds. Notably, AUC-ROC is beneficial for evaluating classifiers on imbalanced datasets as it is unaffected by the class distribution.

### Response
**Interpretation and Analysis of F1 Score and AUC-ROC**

The provided excerpt discusses two crucial performance metrics for classifiers in machine learning: the F1 Score and AUC-ROC. Each has distinct utilities and provides unique insights into a model's performance under different conditions.

**F1 Score: Balancing Precision and Recall**
The F1 Score is the harmonic mean of precision and recall, ensuring that both false positives and false negatives are equally weighted. This metric is essential in situations where both types of errors are costly. For instance, in fraud detection systems, it is critical to minimize both false alarms (false positives) and missed frauds (false negatives). The F1 Score is optimal here as it balances sensitivity (recall) with the precision needed to avoid excessive false alerts.
**AUC-ROC: Evaluating Performance Across Thresholds**
AUC-ROC measures the area under the receiver operating characteristic curve, providing a comprehensive view of model performance across all classification thresholds. This metric is especially valuable in imbalanced datasets, common in medical diagnostics, where the prevalence of one class significantly outweighs the other. AUC-ROC assists in understanding how well a model can distinguish between classes, independent of the class distribution.
**Comparison and Practical Application**
While the F1 Score gives a precise measure of model accuracy at a specific threshold, focusing on the balance between precision and recall, AUC-ROC offers insight into the model's ability to discriminate between classes across thresholds. This makes AUC-ROC suitable for preliminary model analysis and scenarios where class imbalance might skew the perception of model effectiveness provided by other metrics like accuracy.

**Conclusion**
Choosing between F1 Score and AUC-ROC, or using them in conjunction, depends on the specific needs of the classification task and the cost implications of errors. Understanding these metrics' implications can significantly enhance the quality and depth of your review article, providing a clearer guide for model evaluation in diverse practical scenarios. Both metrics together provide a robust framework for evaluating classifier performance, especially in complex real-world settings.

### Query Type: Interpretive Explanation

### Data Generator: gpt-4-turbo-2024-04-09

Figure 7: An example of SkillMix describes a conversation about interpretive explanation and includes two skills, performance metrics analysis and writing review articles.

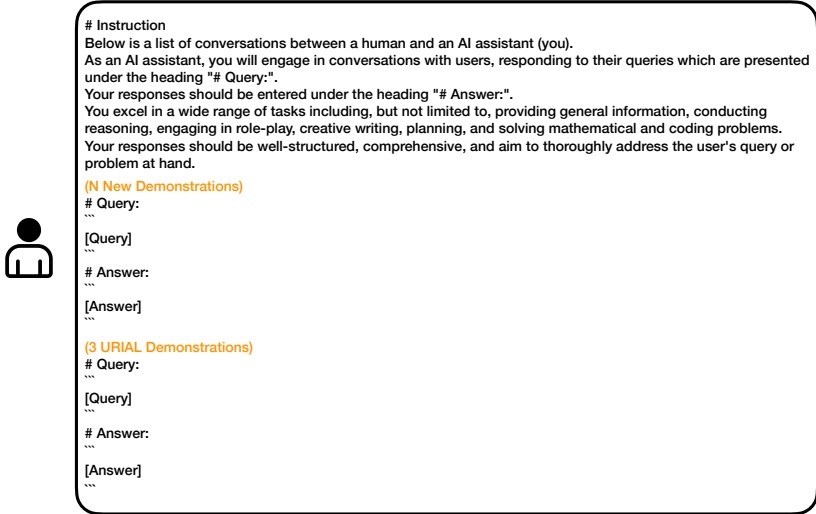

Figure 8: The prompt template for doing ICL in our work. Note that the words in orange are solely for illustration purposes and do not appear in the real prompts. The mixture of new demonstrations we test in our experiments is inserted before URIAL demonstrations.

demonstration, to the original 3-shot URIAL prompt. We evaluate the resulting instruction-following performance for each prompt on MT-Bench but with GPT-4-Turbo as the judge since GPT-4-Turbo is cheaper and faster than GPT-4 (the original LLM judge for MT-Bench evaluation), and also has a high correlation with human judgment. Finally, the example with the highest 1st-turn MT-Bench score is chosen.

We continue our greedy search for the proper fifth and sixth demonstration in a more restricted search space due to the heavy computational cost of the search process. We keep reducing the search space by applying a threshold of 1st-turn MT-Bench score (i.e., 6.2 in our experiments). Similarly, we add each example from the new search space on top of the best (N-1)-shot prompt found in previous steps and generate a set of new N-shot prompts. Then we run the MT-bench evaluation with GPT-4-Turbo as the judge for multiple times and select the best (N+1)-shot prompt.

## A.4 SCALING EXPERIMENTS

The recently released LLMs with increasingly large context windows allow using many more shots for ICL than are typically used. Prior works Agarwal et al. (2024); Bertsch et al. (2024) have shown that many-shot ICL, compared to few-shot ICL, can make task-specific fine-tuning less essential and allows LLMs to tackle a wider range of tasks without specification. Therefore, we want to examine if the instruction-following performance of base LLMs could benefit from many-shot ICL through various scaling experiments, and systematically compare ICL with IFT. In the following, we add more details about these scaling experiments, including the prompt template, data selection for both ICL and IFT, and training hyper-parameters for IFT experiments.

**ICL Random.** We construct the set of in-context demonstrations based on the high-quality data we select from SkillMix-4k or Evol-Instruct-70k dataset. Through randomly sampling from the high-quality dataset for multiple times, we generate a series of in-context demonstration sets that each contains $N$ examples, where $N \in \{0, 7, 17, 27, 37, 47\}$ for Mistral-7B-v0.2 model (32k context length) and $N \in \{0, 7, 17, 27, 37, 47, 97, 147, 197, 247\}$ for Llama-3.1-8B model (128k context length), and insert the $N$ demonstrations into the prompt template for ICL as shown in Fig 8. Note that the prompt template is strictly following the one used in URIAL paper and we only replace the demonstrations used for the ICL purpose. The average performance and standard deviations are computed over 5 random seeds.

```
# Instruction
Below is a list of conversations between a human and an AI assistant (you).
As an AI assistant, you will engage in conversations with users, responding to their queries which are presented under the heading "# Query:".
Your responses should be entered under the heading "# Answer:".
You excel in a wide range of tasks including, but not limited to, providing general information, conducting reasoning, engaging in role-play, creative writing,
planning, and solving mathematical and coding problems.
Your responses should be well-structured, comprehensive, and aim to thoroughly address the user's query or problem at hand.

# Query:
```
Can you tell me some common types of renewable energy sources?

# Answer:
```
Hello! Certainly! Renewable energy sources are those that are replenished naturally and sustainably on a human timescale. Here are some common types of
renewable energy sources:

1. Solar Energy: Energy harnessed from the sun. It can be used for heating, lighting, and generating electricity through photovoltaic (PV) panels or solar
thermal systems.
2. Wind Energy: Energy generated from the natural movement of air. Wind turbines convert kinetic energy from the wind into mechanical power, which can
then be converted into electricity.
3. Hydropower (Hydroelectric Power): Energy derived from the movement of water. This can be the flow of rivers, tidal movements, or wave power.
Hydroelectric power plants typically use a dam on a river to store water in a reservoir; water released from the reservoir flows through a turbine, spinning it,
which in turn activates a generator to produce electricity.
4. Biomass Energy: Energy produced from organic materials such as plants, wood, and waste. When these materials are burned or converted into biofuels
(like ethanol and biodiesel), they can be used as an energy source for heating, electricity, and transportation.
5. Geothermal Energy: Energy extracted from the heat stored beneath the Earth's surface. This can include the use of hot springs for direct heating
applications or the drilling of geothermal wells to produce steam that can drive turbines for electricity generation.

Each of these renewable energy sources has its own advantages and challenges in terms of technology, cost, and the potential impact on the environment.
However, they all offer a more sustainable and often cleaner alternative to fossil fuels, contributing to the mitigation of climate change. Please let me know if
you have any other questions!

# Query:
```
Natalia sold clips to 48 of her friends in April, and then she sold half as many clips in May. How many clips did Natalia sell altogether in April and May?

# Answer:
```
Hi, I'm happy to help you. To solve the problem, let's think step by step:

1. In April, Natalia sold clips to 48 friends.
2. In May, she sold half as many as she did in April. So, the number of clips she sold in May = 48/2 = 24.
3. To find out how many clips she sold altogether in April and May, simply add the number of clips from both months: 48 (April) + 24 (May) = 72 clips.

Therefore, Natalia sold 72 clips altogether in April and May. Please feel free to ask me if you have any more questions!
```

# Query:
```
What is the proper method for deep frying a turkey?
...

# Answer:
```
Hello! I'm happy to help you with your question about deep frying a turkey! Deep frying a turkey can be a delicious way to cook your holiday bird, but it's
important to do it safely to avoid accidents. Here's a step-by-step guide to deep frying a turkey:

1. Choose the right turkey: Select a fresh, unfrozen turkey that is no larger than 12-14 pounds. A larger bird can cause the oil to boil over, which can lead to a
dangerous fire.
2. Prepare the turkey: Thoroughly thaw and dry the turkey before frying. Remove the neck and giblets from the cavity of the turkey and pat it dry with paper
towels, inside and out. This will help prevent oil splatter and ensure a crispy skin.
3. Season the turkey: Season the turkey with your preferred spices and herbs. You can use a dry rub or inject the seasoning directly into the meat.
4. Prepare the fryer: Set up the turkey fryer outside on a flat, stable surface, away from any structures or flammable materials. Fill the fryer with peanut or
canola oil to the 1. recommended level, typically indicated on the fryer. Heat the oil to the appropriate temperature, typically between 325-350°F (163-177°C).
5. Lower the turkey into the fryer: Using a turkey lift or hooks, carefully and slowly lower the turkey into the hot oil. Make sure the turkey is fully submerged in
the oil.
6. Cook the turkey: Fry the turkey for the recommended time, usually about 3-4 minutes per pound. Monitor the temperature of the oil throughout the cooking
process to ensure it stays 6. within the recommended range.
7. Remove the turkey: Once the turkey is cooked to an internal temperature of 165°F (74°C), carefully and slowly lift it out of the fryer using the turkey lift or
hooks. Place it on a wire rack or tray to drain any excess oil.
8. Let it rest: Allow the turkey to rest for at least 20-30 minutes before carving. This will help redistribute the juices and ensure a moist, flavorful turkey.

Remember to always prioritize safety when deep frying a turkey. Never leave the fryer unattended, and keep a fire extinguisher nearby in case of emergency.
Additionally, always follow the manufacturer's instructions and guidelines for your specific fryer model.
```

Figure 9: **The details of URIAL prompt**. It consists of a carefully designed set of rules after "# Instruction" and three curated stylistic examples, each of which always begins with affirming the user query by saying something similar to "Hello! I'm happy to help you.".

**ICL Random (no URIAL).** We generate a series of in-context demonstration sets by randomly sampling from the high-quality IFT dataset, and each set contains $N$ examples, where $N \in \{3, 10, 20, 30, 40, 50\}$ for Mistral-7B-v0.2 model (32k context length) and $N \in \{3, 10, 20, 30, 40, 50, 100, 150, 200, 250\}$ for Llama-3.1-8B model (128k context length). URIAL examples will not be added to the context, so it ensures the total number of in-context examples are the same as the Random group. The average performance and standard deviations are computed over 5 random seeds.

**ICL Greedy Search.** Following the procedure we mention in Sec. A.3, ideally we can get as many optimal examples as we want if we have sufficient OpenAI API credits. Thus it allows us to create another series of in-context demonstration sets by selecting another $N$ examples for each $N \in \{1, 2, 3\}$ in the paper. The resulting mixture of in-context demonstrations is then placed in the corresponding location of the prompt template as shown in Fig. 8.

Table 5: Details of training hyperparameters for IFT experiments.

| Data Size | # GPUs | Epochs | LR | LR Scheduler | Batch Size | Context Win. Len. | WD | Warmup Rate |
|---|---|---|---|---|---|---|---|---|
| *Mistral-7B-v0.2* | | | | | | | | |
| 3 | 2 | 6 | 2e-6 | Cosine | 2 | 2048 | 0.01 | 0.03 |
| 10 | 4 | 6 | 2e-6 | Cosine | 8 | 2048 | 0.01 | 0.03 |
| 20 | 4 | 6 | 2e-6 | Cosine | 8 | 2048 | 0.01 | 0.03 |
| 30 | 4 | 6 | 2e-6 | Cosine | 8 | 2048 | 0.01 | 0.03 |
| 40 | 4 | 6 | 2e-6 | Cosine | 8 | 2048 | 0.01 | 0.03 |
| 50 | 4 | 6 | 2e-6 | Cosine | 8 | 2048 | 0.01 | 0.03 |
| 100 | 4 | 6 | 2e-6 | Cosine | 8 | 2048 | 0.01 | 0.03 |
| 300 | 4 | 6 | 2e-6 | Cosine | 8 | 2048 | 0.01 | 0.03 |
| 1000 | 4 | 15 | 2e-6 | Cosine | 128 | 2048 | 0.01 | 0.03 |
| 2000 | 4 | 15 | 2e-6 | Cosine | 128 | 2048 | 0.01 | 0.03 |
| 4000 | 4 | 15 | 2e-6 | Cosine | 128 | 2048 | 0.01 | 0.03 |
| *Llama-3.1-8B* | | | | | | | | |
| 3 | 2 | 6 | 4e-6 | Cosine | 2 | 2048 | 0.01 | 0.03 |
| 10 | 4 | 6 | 4e-6 | Cosine | 8 | 2048 | 0.01 | 0.03 |
| 20 | 4 | 6 | 4e-6 | Cosine | 8 | 2048 | 0.01 | 0.03 |
| 30 | 4 | 6 | 4e-6 | Cosine | 8 | 2048 | 0.01 | 0.03 |
| 40 | 4 | 6 | 4e-6 | Cosine | 8 | 2048 | 0.01 | 0.03 |
| 50 | 4 | 6 | 4e-6 | Cosine | 8 | 2048 | 0.01 | 0.03 |
| 100 | 4 | 6 | 4e-6 | Cosine | 8 | 2048 | 0.01 | 0.03 |
| 300 | 4 | 6 | 4e-6 | Cosine | 8 | 2048 | 0.01 | 0.03 |
| 1000 | 4 | 15 | 4e-6 | Cosine | 128 | 2048 | 0.01 | 0.03 |
| 2000 | 4 | 15 | 4e-6 | Cosine | 128 | 2048 | 0.01 | 0.03 |
| 4000 | 4 | 15 | 4e-6 | Cosine | 128 | 2048 | 0.01 | 0.03 |

Table 6: **Improved in-context demonstrations selected from Evol-Instruct-70k dataset for Mistral-7B-v0.2 base model.** We report the 1st-turn score on MT-Bench and the length-controlled (LC) win rates on AlpacaEval 2.0 when using different IC prompts.

| Model | MT-Bench (1st) | AlpacaEval 2.0 |
|---|---|---|
| URIAL (3 examples) | 6.99 | 8.09 |
| URIAL + greedy search (1 ex.) | 7.46 | 7.91 |
| URIAL + greedy search (2 ex.) | **7.69** | 8.38 |
| URIAL + greedy search (3 ex.) | 7.68 | 9.22 |

**IFT.** We run instruction fine-tuning the base LLMs (Mistral-7B-v0.2 and Llama-3.1-8B) either with tens of examples (i.e., few-sample regime) or thousands of examples. The training examples are randomly sampled from existing IFT datasets, SkillMix-4k and Evol-Instruct-70k. The mean score and standard deviations are calculated over multiple random seeds. More specifically, we use 5 random seeds for Mistral-7B-v0.2 model and 3 random seeds for Llama-3.1-8B model due to a restricted compute budget. In particular, since the size of SkillMix dataset is 4k, the randomness of IFT with 4k examples primarily comes from optimization process and scoring evaluation with GPT-4, otherwise part of randomness also comes from data sampling process. We list the details of training hyper-parameters used in IFT experiments in Table. 5. We always select the last model checkpoint to run evaluation.

# B REVISIT OUR FINDINGS ON INSTRUCT MODELS

In this section, we want to revisit some of our key findings in the paper on instruct model, including influence of the individual components of URIAL (§ B.1), importance of question-answer matching of demonstrations for in-context alignment (§ B.2), and many-shot in-context alignment (§ B.3). We experiment using Mistral-7B-Instruct-v0.2, the instruct version of Mistral-7B-v0.2 that is widely used in the main paper, and again we adopt MT-Bench with GPT-4 as the judge for evaluation.

Table 7: **Influence of the individual components of URIAL on instruct model** We report the mean MT-Bench scores for cases that have multiple different combinations, such as Rule + 1 example and Rule + 2 example. The standard deviation is always calculated on 3 runs.

| In-context prompt | 1st-turn | 2nd-turn | Average |
|---|---|---|---|
| Empty | **7.92** ± 0.08 | **7.18** ± 0.05 | **7.55** ± 0.06 |
| Rule-only | 7.80 ± 0.07 | 7.15 ± 0.10 | 7.48 ± 0.09 |
| Rule + 1 example | 7.54 ± 0.15 | 6.82 ± 0.34 | 7.18 ± 0.19 |
| Rule + 2 examples | 7.70 ± 0.17 | 6.84 ± 0.21 | 7.27 ± 0.19 |
| URIAL | 7.58 ± 0.06 | 6.48 ± 0.04 | 7.21 ± 0.06 |

Table 8: **Importance of question-answer matching of demonstrations for in-context alignment on instruct models.**

| In-context prompt | 1st-turn | 2nd-turn | Average |
|---|---|---|---|
| no demonstrations | **7.92** | **7.18** | **7.55** |
| $X$ only (no $Y$) | 4.29 | 6.06 | 5.18 |
| $Y$ only (no $X$) | 7.53 | 6.85 | 7.19 |
| circular shift of $Y$ | 6.09 | 5.99 | 6.04 |
| in-domain random $Y$ | 6.62 | 5.94 | 6.28 |
| out-of-domain random | 5.69 | 5.50 | 5.60 |
| correct $Y$ | 7.58 | 6.48 | 7.21 |

## B.1 INFLUENCE OF THE INDIVIDUAL COMPONENTS OF URIAL ON INSTRUCT MODEL

In Table 7, we find that different from what we observed from the base LLM, adding in-context examples, even only rules, weakens the instruction-following performance of instruct models as measured by MT-Bench performance, and more examples lead to more performance declines.

## B.2 IMPORTANCE OF QUESTION-ANSWER MATCHING OF DEMONSTRATIONS FOR IN-CONTEXT ALIGNMENT ON INSTRUCT MODELS

In Table 8, we find that adding examples in the context consistently causes damage to the instruction-following performance of instruct models, even when using correct URIAL examples, which is similar to what we observe on the instruct model in Table 7. Surprisingly, it leads to a pretty bad 1st-turn score (4.29) on MT-Bench when only presenting queries and no corresponding answers in the context. After looking into the model responses, we find that the instruction model sometimes would manage to answer all queries present in the context, including the ones as demonstrations, and outputs consisting of answers to multiple questions lead to low performance on model-based automatic evaluations. This unexpected behavior is mitigated in the 2nd-turn evaluation as a complete query-answer pair is added to the context after the 1st-turn answer generation.

## B.3 MANY-SHOT IN-CONTEXT ALIGNMENT ON INSTRUCT MODEL

We further test the effect of alignment with ICL on Mistral-7B-Instruct-v0.2 when increasing the number of demonstrations. We test on the same data we used in Fig. 3 and more details can be found in App. A.4. The results of many-shot in-context alignment experiments are shown in Fig. 10. We find that alignment through ICL does not further improve the instruction-following performance of instruct models, but slightly declines its capability to answer user queries.

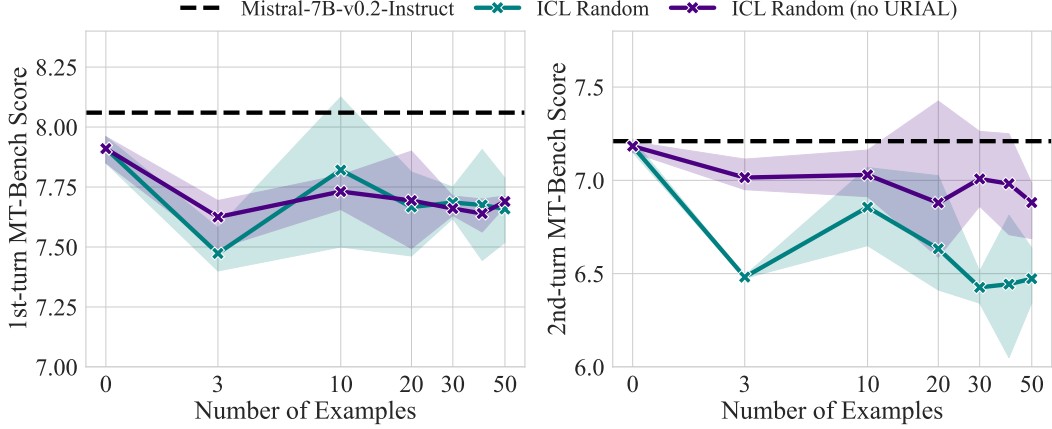

Figure 10: **Scaling the number of demonstrations for alignment with ICL on Mistral-7B-Instruct-v0.2**. We measure the alignment performance of different settings using the MT-Bench score. The base model is always Mistral-7B-Instruct-v0.2 for ICL Random and ICL Random (no URIAL).

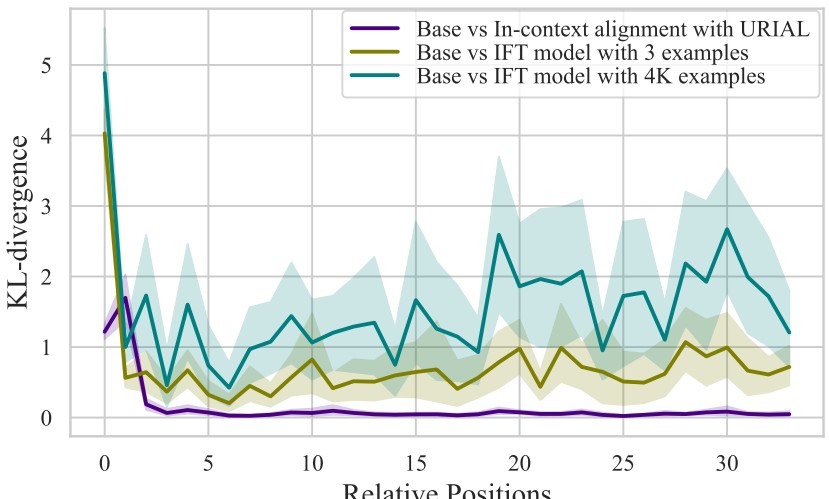

Figure 11: **KL-divergence between the base LLM and its aligned model using ICL or IFT.** The base model is always Mistral-7B-v0.2. The KL-divergence results are averaged over 25 examples.

## C    KL-DIVERGENCE BETWEEN THE BASE LLM AND ITS ALIGNED MODEL USING ICL OR IFT

Since we always experiment with open source LLMs with $7 \sim 8$ billions parameters, we are interested in mechanistic study with logits of model outputs to reveal more insights, such as the KL-divergence between the base LLM and its aligned counterparts.

**Experimental setup.** We randomly select 25 query-response paired examples from SkillMix dataset as the data to calculate KL-divergence between the base LLM and the aligned models. Specifically, for each query-response paired example, two models predict the next token given the same input respectively. The model only predicts tokens from responses and each answer token after calculating KL-divergence will be added to the input for the next prediction. The response from each example is truncated to save computation costs. We always select Mistral-7B-v0.2 as the base model, and another 3 aligned models, in-context alignment using URIAL, IFT with 3 SkillMix examples, and IFT with 4k SkillMix examples, as target models to calculate KL-divergence.

**Results.**    From Fig. 11, we observe that the probability distribution of the next token prediction, for the in-context aligned model, differs significantly only for the first few tokens, after which the

Table 9: **Transferability of greedy search IC prompt to other LLMs.** We use the 1st-turn score on MT-Bench and the length-controlled (LC) win rates on AlpacaEval (AE) 2.0 when using various IC prompts on different base models.

| Model | MT-B (1st) | AE 2.0 |
|---|---|---|
| **Base model: Llama-2-7B-80k** | | |
| URIAL (3 examples) | 5.19 | 1.81 |
| URIAL + greedy search (1 ex.) | 5.56 | 1.50 |
| URIAL + greedy search (2 ex.) | **5.57** | 1.84 |
| URIAL + greedy search (3 ex.) | 5.10 | **2.91** |
| **Base model: Mixtral-8x22B-v0.1-4bit** | | |
| URIAL (3 examples) | 8.28 | 14.77 |
| URIAL + greedy search (1 ex.) | 8.36 | **15.74** |
| URIAL + greedy search (2 ex.) | 7.79 | 13.20 |
| URIAL + greedy search (3 ex.) | **8.58** | 14.68 |

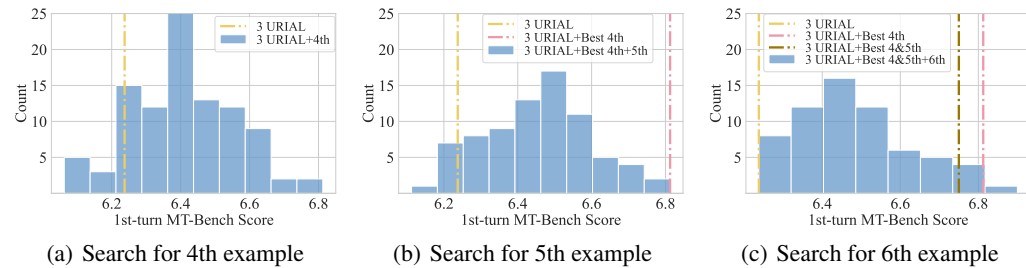

(a) Search for 4th example     (b) Search for 5th example     (c) Search for 6th example

Figure 12: **The distribution of the 1st-turn MT-Bench score (GPT-4-Turbo as judge) on Mistral-7B-v0.2 obtained by adding multiple instructions from SkillMix (Kaur et al., 2024) as a 4th (a); 5th (b); 6th (c) demonstration to URIAL.** The dashed lines of various colors refer to the 1st-turn MT-Bench score of the obtained searching results. A majority of the 4th examples have positive contributions to the model's instruction-following performance, but the improvement quickly diminishes when running the greedy search for more optimal demonstrations.

KL-divergence value rapidly drops to approximately 0. However, the KL-divergence between the IFT models and the base model does not converge to 0 and has larger KL-divergence values on average. Surprisingly, IFT on only 3 examples is already sufficient to drive the fin-tuned model far away from the base model as measured by KL-divergence compared to in-context alignment. In particular, the difference in the probability distribution of next token prediction becomes more substantial as the base model are instruction fine-tuned on more data.

## D    TRANSFERABILITY OF IN-CONTEXT PROMPTS

In Table 9, we report the performance of applying the in-context examples found by greedy search (added to URIAL) on Mistral-7B-v0.2 and Evol-Instruct-70k dataset (see Table. 6) to Llama-2-7B-80k (Fu et al., 2024) and Mixtral-8x22B-v0.1-4bit (Jiang et al., 2024). Adding the new examples does not provide a consistent improvement: while it can increase the MT-Bench score (+0.47 on Llama-2-7B-80k, +0.30 on Mixtral-8x22B-v0.1-4bit), it can also, in some cases, give worse results than the original URIAL. Similarly, it yields mixed results when measured by win rate on AlpacaEval 2.0, which was not optimized by the greedy search. This analysis suggests that the most effective ICL demonstrations might vary across base LLMs, potentially because of differences in their pre-training. This could further explain why URIAL underperforms on multiple models in Table 1, especially on those which have become available only recently and were not used for selecting the URIAL examples.

Table 10: **Scaling experiment results (length-controlled win-rates) on AlpacaEval 2.0.** We put different amounts of random examples drawn from SkillMix to the context as demonstrations. We conduct experiments on two base models: Mistral-7B-v0.2 and Meta-Llama-3.1-8B. The results are computed over 3 different runs.

| | AlpacaEval 2.0 LC Win-rate (%) | |
| --- | --- | --- |
| | Mistral-7B-v0.2 | Meta-Llama-3.1-8B |
| 0 example | $2.92 \pm 0.0$ | $2.99 \pm 0.0$ |
| 1 example | $11.98 \pm 0.7$ | $8.66 \pm 2.7$ |
| 3 examples | $12.50 \pm 1.3$ | $11.56 \pm 0.6$ |
| 10 examples | $14.86 \pm 0.4$ | $15.11 \pm 1.9$ |
| 20 examples | $13.20 \pm 0.7$ | $15.18 \pm 0.8$ |
| 30 examples | $12.45 \pm 0.1$ | $13.90 \pm 0.2$ |
| 40 examples | $12.51 \pm 1.3$ | $14.95 \pm 0.7$ |
| 50 examples | $13.80 \pm 0.7$ | $15.49 \pm 0.8$ |

## E   ADDITIONAL EXPERIMENTS

### E.1   SCALING EXPERIMENTS ON ALPACAEVAL 2.0

This section provides additional results of scaling experiments conducted on the AlpacaEval Suite, which contains 805 single-turn examples, an order of magnitude more than MT-Bench. Based on the SkillMix dataset, we construct the set of in-context demonstrations by sampling different amounts of random examples, ranging from 0 to 50 examples. The results shown in Table 10 confirm our findings, reported in Sec. 3, that scaling the number of ICL examples is not sufficient to consistently improve the instruction following performance.

### E.2   GREEDY SEARCH ON MISTRAL-7B-V0.2

In this section, we show the distribution of the resulting scores (with GPT-4-Turbo as the judge) for each step of the greedy search on SkillMix-4k dataset when using Mistral-7B-v0.2 as the base model. Similar to the finding on Llama-3.1-8B model (see Sec. 3.2), the majority of demonstrations increase the alignment performance, as measured by 1st-turn MT-Bench score, when added as the fourth example on top of URIAL, and few progress is made with more demonstrations. We show more details of greedy search results on Mistral-7B-v0.2 in Fig. 12.

### E.3   MORE EXPERIMENTS ON DECODING SCHEMES

We show more results of decoding schemes under different settings in this section, as supplementary to the main results in Sec. 2.2. Specifically, in Fig. 13, we turn off the repetition penalty (i.e., repetition penalty = 1.0) and show the 1st-turn MT-Bench scores on Mistral-7B-v0.2 when varying the values of temperature and top-p. Moreover, we repeat the same experiment procedure but on a different base model, Llama-3-8B, and show heatmaps in Fig. 14.

### E.4   A BREAKDOWN EXAMINATION OF URIAL-ALIGNED MODELS

The test set of MT-Bench (Zheng et al., 2023) is comprised of high-quality questions that belong to 8 common categories: coding, math, reasoning, extraction, roleplay, writing, humanities/social science, and STEM. In Table 11, we present the per-category performance on MT-Bench for each model in order to have a more detailed comparison between the URIAL-aligned models and the corresponding instruct models. The results suggest fine-tuning is almost always better than URIAL (ICL) and only underperforms in a minority of cases on particular base models, such as Mistral-7B-v0.1 and Llama-2-70B.

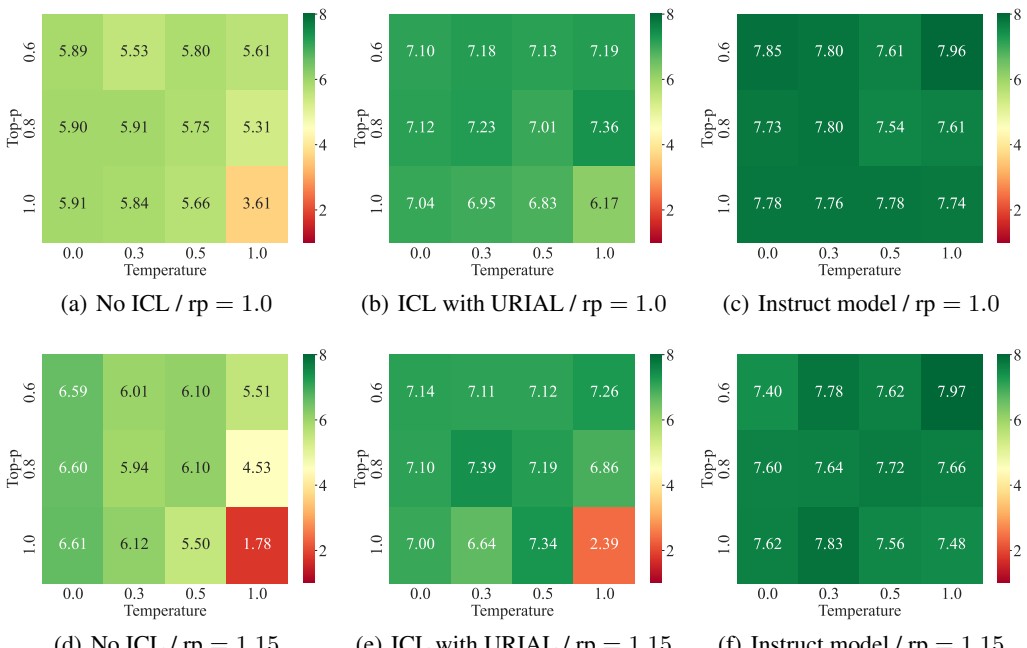

Figure 13: **The 1st-turn MT-Bench scores of model generations with and without ICL across different decoding schemes.** We mainly consider two hyper-parameters: temperature and top-p. The heatmaps show the answering quality of the base model with and without URIAL in the context is sensitive to the setups of decoding schemes. Surprisingly, with proper decoding parameters, the base model alone is already capable of following instructions.

Table 11: **A breakdown examination of URIAL-aligned models and corresponding instruct models across different categories on MT-Bench.** * denotes the results taken from the URIAL GitHub repository.

| Model | coding | math | reasoning | extraction | humanities | roleplay | stem | writing | Average |
|---|---|---|---|---|---|---|---|---|---|
| Llama-2-7B + URIAL * | 1.65 | 1.60 | 3.45 | 3.40 | 8.08 | 7.48 | 6.80 | 6.20 | 4.83 |
| Llama-2-7B-Instruct | **2.95** | **2.40** | **5.20** | **6.33** | **9.58** | **7.83** | **8.88** | **9.05** | **6.53** |
| Llama-2-70B + URIAL * | **4.15** | 3.60 | **6.10** | **7.70** | 9.75 | 7.33 | 8.75 | **9.50** | 7.11 |
| Llama-2-70B-Instruct | 3.75 | **4.10** | 5.95 | 7.40 | **9.85** | **7.90** | **9.13** | 9.50 | **7.20** |
| Llama-3-8B + URIAL * | 4.15 | 2.60 | 3.50 | 5.25 | 8.90 | 7.30 | 8.15 | 6.13 | 5.75 |
| Llama-3-8B-Instruct | **5.95** | **5.05** | **6.15** | **9.16** | **9.90** | **9.05** | **8.95** | **8.70** | **7.86** |
| Llama-3-70B + URIAL * | 4.35 | 3.80 | 4.95 | 6.20 | 8.00 | 7.25 | 8.55 | 8.10 | 6.40 |
| Llama-3-70B-Instruct | **7.85** | **7.35** | **6.25** | **9.75** | **10.00** | **9.30** | **9.60** | **9.80** | **8.74** |
| Llama-3.1-8B + URIAL * | 4.35 | 3.25 | 3.95 | 5.95 | 9.00 | 6.95 | 8.00 | 7.60 | 6.13 |
| Llama-3.1-8B-Instruct | **6.40** | **6.50** | **5.70** | **8.78** | **9.80** | **9.00** | **8.60** | **9.20** | **8.00** |
| Mistral-7B-v0.1 + URIAL * | **4.60** | 3.40 | 4.90 | **7.75** | 9.08 | **7.65** | **8.28** | 7.75 | 6.67 |
| Mistral-7B-Instruct-v0.1 | 4.35 | **3.95** | **6.30** | 6.75 | **9.45** | 7.45 | 7.70 | **8.85** | **6.85** |
| Mistral-7B-v0.2 + URIAL * | 3.80 | 3.35 | 4.50 | 7.45 | 8.95 | 6.70 | 7.43 | 7.98 | 6.27 |
| Mistral-7B-Instruct-v0.2 | **5.45** | **3.40** | **6.50** | **8.50** | **9.90** | **8.65** | **9.30** | **9.40** | **7.64** |
| Mixtral-8x22B-v0.1-4bit + URIAL | 5.25 | 5.85 | 6.00 | **9.20** | **9.80** | 8.65 | 8.30 | 8.60 | 7.71 |
| Mixtral-8x22B-Instruct-v0.1-4bit | **7.10** | **7.03** | **7.00** | 9.10 | 9.65 | **8.90** | **9.65** | **9.70** | **8.52** |
| GPT-4-Base + URIAL | 5.55 | 5.98 | 6.90 | 8.20 | 5.93 | 6.90 | 6.45 | 6.10 | 6.50 |
| GPT-4 (March 2023) * | **8.55** | **6.80** | **9.00** | **9.38** | **9.95** | **8.90** | **9.70** | **9.65** | **8.99** |

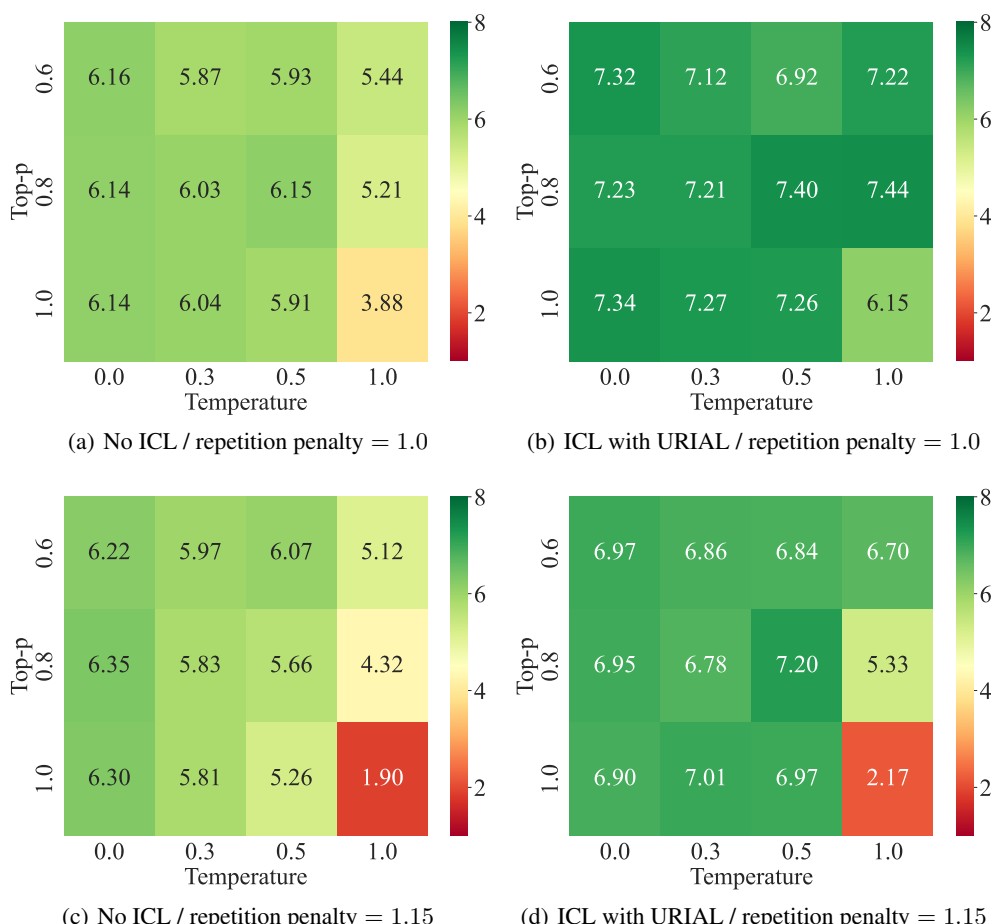

Figure 14: **The 1st-turn MT-Bench scores of Llama-3.1-8B generations with and without URIAL in the context across different decoding schemes.** We mainly consider two hyper-parameters: temperature and top-p.

