# OpenReview forum: "Is In-Context Learning Sufficient for Instruction Following in LLMs?"
_ICLR.cc/2025/Conference — ICLR 2025 Poster_

### Official Review · Reviewer_rg7y · 2024-10-28

**Soundness:** 3
**Presentation:** 2
**Contribution:** 3
**Rating:** 6
**Confidence:** 3

**Summary:**

This paper continues on URIAL work to deep dive into the comparison of ICL vs IFT. It revealed crucial insights to improve ICL performance, i.e. high quality examples and temperate as decoding parameters. On the flip side, it also shows the shortcoming of ICL, especially on subsequent turn in conversational benchmarks.

**Strengths:**

This paper contributes valuable guidance to practitioners who often struggle between the choice of finetuning and prompting. The work is particularly useful for scenarios where obtaining training data is challenging. Also when all attentions are on instruct models, this paper shows that base models alone can be instructed and aligned, and made useful. This shall open the gate to lots of research projects too.

**Weaknesses:**

More attribution works are needed to answer why the outcome is what it is. For example, the poor performance of the second-turn in MT benchmark definitely needs deeper dive. Is it because of the formatting or lack of examples? What caused the ICL to lose grip of the first turn?

Considering that the authors work on small size open LLMs (except GPT-4), mechanistic study with logits or internal states shall definitely reveal more insights.

**Questions:**

For prompts ending up with varying output performances, what are their KL divergence?

If you repeat your experiments on instruct models instead of base models, what would the results be?

How does ICL cope with long-context tasks (summarization, coding, etc.) where context window limit the extent of examples?

---

> ### Author Response · Authors · 2024-11-25
> **Official Comment by Authors**
>
> We thank the reviewer for the detailed comments and for highlighting the relevance of our work. In the following, we address the raised questions.
> - **The poor performance of the second-turn in MT benchmark definitely needs deeper dive. Is it because of the formatting or lack of examples?**
>
>     Thanks for your suggestions.  We explored multi-turn alignment via ICL on two open-source LLMs, Mistral-7B-v0.2 and Llama-3.1-8B. We also ablated factors such as the number of multi-turn conversations and the format of in-context demonstrations. We show the results in Section 2.4 (Line 269-307) in the revised version. Table 3 indicates that including multi-turn examples in the context consistently improves the model’s capability of handling multi-turn conversations, as measured by 2nd-turn MT-Bench score. For example, using a well-structured context  (clearly separating single-turn and multi-turn examples into distinct groups), improves the 2nd-turn performance, from 5.88 to 6.31 on Mistral-7B-v0.2 and from 6.43 to 6.71 on Llama-3.1-8B. However, increasing the number of multi-turn examples in the context does not provide additional benefits.
> - **What caused the ICL to lose grip of the first turn?**
>
>     We believe the primary reason is the fact that the original URIAL used only single-turn examples. However, as our new results imply, we can directly improve the 2nd-turn performance if we include two-turn examples in-context. Note, however, that there is a curious difference between ICL and IFT: for IFT fine-tuning on single-turn conversations does lead to generalization to multi-turn ones. This highlights an interesting conceptual difference between ICL and IFT.
> - **Considering that the authors work on small size open LLMs (except GPT-4), mechanistic study with logits or internal states shall definitely reveal more insights.**
>
>     Thanks for your suggestion. We computed the KL-divergence between the base LLM and its aligned counterparts, where alignment was performed using either ICL or IFT, and added these results in Appendix C (Line 1064-1120). We will add more mechanistic studies involving logits and internal states in the final version to reveal additional insights. We comment on the findings of this experiment below.
> - **For prompts ending up with varying output performances, what are their KL divergence?**
>
>     As explained in the previous reply, to address this question, we calculated the KL-divergence between the base LLM and its aligned counterparts via ICL or IFT, and presented the results in Appendix C (Line 1064-1120). As shown in Figure 11, we observe that the probability distribution of next-token predictions for the in-context aligned model differs significantly only for the first few tokens, after which the KL-divergence rapidly drops to approximately 0. However, the KL-divergence between the IFT models and the base model does not converge to 0 and has larger KL-divergence values on average. Surprisingly, IFT on only 3 examples is already sufficient to drive the fine-tuned model far away from the base model as measured by KL-divergence compared to in-context alignment. In particular, the difference in the probability distribution of the next token prediction becomes more substantial as the base model is instruction fine-tuned on more data.
> - **If you repeat your experiments on instruct models instead of base models, what would the results be?**
>
>     Following your suggestions, we have added new results on instruct models, covering the following experiments:
>
>     - **Influence of the individual components of URIAL on Instruct models (App. B.1)**. In Table 7, we observe that, unlike for the base LLM, adding in-context examples, even just rules, weakens the instruction-following performance of instruct models, as measured by MT-Bench. Moreover, increasing the number of examples exacerbates the performance decline.
>     - **Importance of question-answer matching of demonstrations for in-context alignment on Instruct models (App. B.2)**. In Table 8, we show that adding examples in the context consistently declines the instruction-following performance of instruct models, even when using correct URIAL examples.
>     - **Many-shot in-context alignment on Instruct models (App. B.3)**. The results in Figure 10 suggest that further alignment through ICL, even with more in-context demonstrations, does not improve the instruction-following performance of instruct models.
>
>     We think that these evaluations complements our previous results, and provides an even more comprehensive understanding of the ICL for instruction following alignment.

---

> ### Author Response · Authors · 2024-11-25
> **Official Comment by Authors (2/2)**
>
> - **How does ICL cope with long-context tasks (summarization, coding, etc.) where context window limit the extent of examples?**
>
>     Thank you for your comment. Our work focuses on aligning base LLMs for instruction-following tasks. For insights into the performance of ICL on long-context tasks such as summarization, machine translation, planning, problem-solving, and question-answering, we refer to Many-shot In-Context Learning[https://arxiv.org/abs/2404.11018].
>
> We thank again the reviewer for the feedback. We hope our rebuttal addresses the raised concerns.

---

> ### Author Response · Authors · 2024-11-29
> **Have our responses addressed your concerns?**
>
> Dear Reviewer rg7y,
>
> Thank you again for your review and comments. Since the discussion period is closing soon, we are wondering whether our responses have addressed your concerns. If not, we would be happy to continue the discussion and provide further clarifications or explanations.

---

> ### Author Response · Authors · 2024-12-01
> **Follow-Up**
>
> Dear Reviewer rg7y,
>
> We would like to follow up on our discussions and check if our previous response has addressed your concerns.
>
> To briefly recap:
>
> - We studied the impact of **multi-turn examples** on in-context alignment, taking into account formatting and the quantity of examples (Section 2.4 and Table 3).
> - We revisited three primary experiments, originally performed on base LLMs, with **instruct models** (Appendix B.1, B.2, B.3 and Figure 7, 8, 10).
> - We computed **KL-divergence** between the base LLMs and its aligned counterparts via ICL or IFT (Appendix C and Figure 11).
>
> Could you please let us know if these updates properly answer your questions and address your concerns about additional attribution works to understand the outcome?
>
> We are willing to address any further concerns before the discussion period ends. Thanks again for your time and valuable feedback!

---

> > ### Comment · Reviewer_rg7y · 2024-12-02
> >
> > Thank you for the additional work. Score 7 does not exist, and I don't think this paper is up to the score 8. I think comments of reviewer GajX comments should be your focus until the final deadline.

---

### Official Review · Reviewer_GajX · 2024-11-02

**Soundness:** 3
**Presentation:** 2
**Contribution:** 1
**Rating:** 3
**Confidence:** 3

**Summary:**

An interesting empirical study of the existing URIAL method that allows alignment via ICL. Unclear whether the contributions are significant.

**Strengths:**

The paper conducts a comprehensive empirical analysis of the URAIL method for aligning pre-trained LLMs via in-context learning.
It considers different datasets, LLMs and test scenarios of relevance.
The figures are well done and show interesting trends. The paper is well structured and easy to follow.

**Weaknesses:**

I am unsure about the exact contributions of this work. It appears mainly as an extended study of the URIAL method on (a) additional models, (b) additional datasets, and (c) more hyperparameters. A relevant contribution of the work is the greedy ICL prompt search, which shows significant performance improvements (i.e., higher performance with much fewer ICL prompts, i.e., with much fewer test-time compute requirements). However, the entire analysis is again primarily based on a single dataset (MT bench). The language sometimes could be more academic, consider e.g. “suffers from some heavy overfitting”.

**Questions:**

- What do the authors see as the work's main contribution, apart from the empirical study of a previously introduced method?
- See other aspects mentioned above

---

> ### Author Response · Authors · 2024-11-25
> **Official Comment by Authors**
>
> We thank the reviewer for the detailed comments. In the following, we address them in details.
> - **I am unsure about the exact contributions of this work. It appears mainly as an extended study of the URIAL method on (a) additional models, (b) additional datasets, and (c) more hyperparameters.**
>
>     Thanks for your comment. We have added a summary of our main contributions at the end of the introduction section (Line 110-123) which we present below:
>
>     - **Analysis of In-Context Alignment**: We systematically evaluate URIAL on a broader set of base LLMs, including GPT-4-Base, with established instruction-following benchmark. Our results indicate that **in-context alignment with URIAL still underperforms instruction fine-tuning** and more sophisticated alignment methods (unlike what suggested by Lin et al. (2024)[https://arxiv.org/abs/2312.01552]), and a proper decoding scheme is the crucial ingredient behind the empirical success of URIAL, which was not throughly discussed in Lin et al. (2024)[https://arxiv.org/abs/2312.01552]
>     - **Scaling In-Context Alignment**: We find that many-shot ICL with high-quality examples and a simple greedy algorithm can reduce the gap between in-context aligned models to fine-tuned models, which had not been explored by prior works.
>     - **First systematic comparison of ICL vs IFT for instruction following**: We show that ICL and IFT with the same number of examples are roughly equivalent for single-turn conversations in the low-data regime. However, IFT generalizes substantially better than ICL when more examples are present, especially for multi-turn conversations.
>
>     Therefore, our work is not simply an extended study of the URIAL method on additional models, datasets, and hyper-parameters. Instead, it provides valuable insights into the fundamental differences and similarities between in-context learning and instruction fine-tuning.
> - **The entire analysis is again primarily based on a single dataset (MT bench).**
>
>     Thanks for your comment. We also adopt AlpacaEval 2.0 evaluation benchmark in Table 4 in our paper. We primarily use the established MT-Bench evaluation for two reasons: (a) it allows us to track instruction-following performance on first- and second-turn conversations; and (b) we want to use an evaluation that is consistent across all experiments in the paper, and MT-Bench is a good choice for balancing evaluation accuracy with practical constraints, such as time and cost.
> - **The language sometimes could be more academic, consider e.g. “suffers from some heavy overfitting”.**
>
>     Thanks for your suggestion for improving the writing. We have improved our language by using “ICL overfits to the style” (Line 104).
> - **What do the authors see as the work's main contribution, apart from the empirical study of a previously introduced method?**
>
>     Thanks for your comment. We have restated our contributions in the previous question, and we want to highlight again here. Our work revisits in-context alignment on a broader set of base LLMs, including GPT-4-Base, finding that decoding parameters are key ingredients behind the empirical success of URIAL. We also show that in-context alignment using only the URIAL prompt cannot close the gap to the aligned counterparts achieved with more sophisticated alignment approaches. Besides, we study methods to improve upon URIAL by using many-shot ICL and a simple greedy algorithm, further revealing the critical role of data quality. Finally,  to the best of our knowledge, ours is the first work to systematically compare ICL and instruction fine-tuning (IFT) for instruction following in the low data regime.
>
>     In the revised version, we have included new results on (a) multi-turn alignment via ICL (Section 2.4, Line 267-308); (b) Analysis of in-context alignment on instruct models (Appendix B, Line 965-1025); (c) KL-divergence between the base LLM and its aligned counterparts via ICL or IFT (Appendix C, Line 1064-1120).
>
> We thank again the reviewer for the feedback. We hope our rebuttal addresses the raised concerns.

---

> > ### Comment · Reviewer_GajX · 2024-11-27
> >
> > Dear authors, thank you for the detailed response, which addresses some of my concerns. However, I believe that the choice and diversity of the used data is of critical importance for an empirical analysis.
> > Upon further inspection of the MT-Bench dataset, it appears that this dataset is only composed of 80 samples. The vast part of the analysis is based on this dataset (i.e. 7 Figures / Tables in the paper are largely based on results from it.) I do not believe that this one dataset is sufficient for the conclusions drawn in the paper, and updated my score accordingly.

---

> > > ### Author Response · Authors · 2024-11-27
> > > **Thanks for the follow-up comment and your previous feedback**
> > >
> > > Thanks for the follow-up comment and your previous feedback!
> > >
> > > It's true that MT-Bench has 80 first-turn and 80 second-turn samples. However, there are multiple reasons why we think our evaluation setting is meaningful:
> > > - [MT-Bench](https://arxiv.org/abs/2306.05685) (published at NeurIPS 2023) is a widely cited and used dataset.
> > > - MT-Bench is, in fact, composed of *diverse* categories, such as coding, math, reasoning, humanities, writing, etc.
> > > - We replicate most of our results over 5 random seeds, which makes our results and findings more reliable. As a result, we observe clear trends, even with 80 samples.
> > > - We cross-check our findings on AlpacaEval 2.0 (LC) for a selected subset of experiments.
> > > - We see a consistent trend across different LLMs—this confirms that MT-Bench is sufficient to reveal systematic signal and not just noise.
> > > - Even 80 prompts of MT-Bench already leads to pretty expensive evaluations due to the usage of GPT-4 as a judge. We spent in total $\approx$\$4'000 on the experiments for this paper. Doing similar experiments on a dataset that is an order of magnitude larger would be prohibitively expensive for us.

---

> ### Author Response · Authors · 2024-11-29
> **Have our responses addressed your concerns?**
>
> Dear reviewer GajX,
>
> Thank you again for your review and comments. I am writing to emphasize the importance of your review for our submission. The discussion period is closing soon, and we are wondering whether our responses have addressed your concerns, particularly with the main evaluation benchmark used in the paper. If not, we would be happy to continue the discussion and provide further clarifications or explanations. Your expertise is highly valued, and we trust that a reconsidered review will reflect the true merit of our work.

---

> > ### Comment · Reviewer_GajX · 2024-12-02
> >
> > Dear authors, thank you for the clarifications.
> > Some thoughts below:
> > - Why was it not possible to run further experiments on the AlpacaEval Suite?
> > - Why were no other multi-turn question answering datasets considered for this work? Specifically, which ones were considered and why were they deemed not suitable?
> > - I understand that financial limitations may prohibit GPT4 evals on large datasets. However this is a limitation that also applied to other academic works, which anyhow managed to run their experiments on more than 80 samples x 5 seeds.

---

> ### Author Response · Authors · 2024-12-04
> **Further reply to Reviewer GajX**
>
> Thank you for your thoughtful feedback.
> - **Why was it not possible to run further experiments on the AlpacaEval Suite?**
>
>     Thank you for the suggestion. We are ready to add these experiments. AlpacaEval contains 805 **single-turn** examples (i.e., an order of magnitude more than MT-Bench), but it uses GPT-4 Turbo as the LLM judge instead of the original GPT-4, which makes the experiments less expensive. Given the limited remaining time in the discussion phase, we provide preliminary results below and commit to including the results for the main scaling experiments in the final version of the paper. The table below shows the results of in-context alignment for different numbers of randomly selected examples from the Skill-Mix dataset (Mistral-7B-v0.2 as base model). We compute the evaluation results based on 3 different runs.
>
>     This evaluation on AlpacaEval shows a trend consistent with the observations on MT-Bench: a few in-context examples are sufficient to significantly improve instruction-following performance, but further increasing their number does not provide additional gains. These results indicate that our observations generalize to different instruction following datasets, without overfitting to the MT-Bench evaluation.
>     |  | AlpacaEval 2.0 LC Win-rate (%) |
>     | --- | --- |
>     | 0 example | $2.92 \pm 0.0$ |
>     | 1 example | $11.98 \pm 0.7$ |
>     | 3 examples  | $12.50 \pm 1.3$ |
>     | 10 examples  | $14.86 \pm 0.4$ |
>     | 30 examples | $12.45 \pm 0.1$ |
>     | 50 examples | $13.80 \pm 0.7$ |
> - **Why were no other multi-turn question answering datasets considered for this work? Specifically, which ones were considered and why were they deemed not suitable?**
>
>     We note that studying performance of ICL alignment in the multi-turn setting was not the main focus of our original submission. However, we agree with the reviewers that this is an important question, and in fact we have added additional experiments in this direction during the rebuttal (see the [general response](https://openreview.net/forum?id=STEEDDv3zI&noteId=sp4RL4DuDf)).
>
>     We would be happy to expand the evaluation on additional multi-turn datasets in the final version of the paper. For example, the two most promising alternative benchmarks with evaluation on multi-turn conversations which we may include are [MT-Eval](https://arxiv.org/abs/2401.16745) and [WildBench](https://arxiv.org/abs/2406.04770). We would also appreciate it if the reviewer could point to any additional suitable multi-turn benchmarks that we may have missed.
> - **I understand that financial limitations may prohibit GPT4 evals on large datasets. However this is a limitation that also applied to other academic works, which anyhow managed to run their experiments on more than 80 samples x 5 seeds.**
>
>     Thank you for your comment. Since MT-Bench is widely used in both academic paper and technical reports from industry, we chose to stick to the exact MT-Bench evaluation protocol, which includes the original GPT-4 as a judge. Alternatively, we can use GPT-4 Turbo or GPT-4o as judges at a lower price, and we can rerun our main evaluations on larger datasets, such as MT-Eval or WildBench.

---

### Official Review · Reviewer_ZLSn · 2024-11-04

**Soundness:** 2
**Presentation:** 2
**Contribution:** 2
**Rating:** 8
**Confidence:** 3

**Summary:**

This paper studies whether in-context learning can be competitive with instruction fine-tuning methods on instruction following tasks. The empirical results ablate many different components of in-context learning methods and the dataset composition. The results are comprehensive and yield several potentially useful results (e.g., how decoding parameters impact the performance). The overall conclusion is that in-context learning is worse than and not as scalable as instruction fine-tuning.

**Strengths:**

The experimental results are comprehensive. The problem studied in this paper is of interest to the deep learning community today.

**Weaknesses:**

Writing is not clear. See below:
- The author should have a brief introduction to URIAL since it's heavily used in the later sections when explaining the experimental results. The current version refers to URIAL's components with the assumption that the readers know URIAL in detail. The current explanation like Line 120-126 is not enough. What do you mean by stylistic examples? What do you mean by rules? What do you mean by "begin with affirming the user queries?" I suggest adding some examples in the main paper to help the readers understand the idea.
- Similarly, what SkillMix datasets do has to be explained.
- This paper presents many experimental findings in each section, but the messages of these findings were not highlighted. For example, Section 2.1 has the title "Systematic Evaluation of URIAL" and starts with a brief explanation of URIAL. When reading this section, I couldn't expect what you will be talking about in this section and I quickly get lost while reading a bunch of implementation details. Even after finishing the section, I couldn't get the takeaway message clearly. Section 2.2, in contrast, did a better job on explaining the results.

**Questions:**

- The abbreviation URAIL is not defined.
- Line 139: If providing multi-turn examples improves performance at multi-turn conversation, why do the authors not study it in this paper?
- The results in Figure 2 are not representative in my opinion since the confidence bounds are overlapped largely.
- Line 320: You say including examples from SkillMix is better than including examples from URIAL. Does URIAL come with a curated dataset as well?
- Figure 3: I don't get it clearly. Is "Llama-3.1-8B-instruct" a model with instruction fine-tuning? Are these rest of ICL methods based on Llama-3.1-8B base models?

---

> ### Author Response · Authors · 2024-11-25
> **Official Comment by Authors**
>
> We thank the reviewer for the detailed comments and for appreciating the relevance of our work. In the following, we address the identified weaknesses and questions.
> - **The current version refers to URIAL's components with the assumption that the readers know URIAL in detail. The current explanation like Line 120-126 is not enough. What do you mean by stylistic examples? What do you mean by rules? What do you mean by "begin with affirming the user queries?" I suggest adding some examples in the main paper to help the readers understand the idea.**
>
>     Thanks for your comments and suggestions for improving our writing. We have updated the description of URIAL in Lines 135-144 and added more details to explain what stylistic examples, rules, and “begin with affirming the user queries” are. We have also added the complete URIAL prompt in the Appendix (Fig. 9).
> - **Similarly, what SkillMix datasets do has to be explained.**
>
>     Following your suggestion, we have added a paragraph (Line 319-323) to introduce the SkillMix dataset at the beginning of Section 3. Besides we add two examples of the SkillMix dataset in the appendix (Fig. 6 and Fig. 7, Line 702-723 and Line 756-777) that also include some metadata like skills, query type, and data generator.
> - **The messages of findings in Section 2.1 are not highlighted. This section is poorly organized and easy to get lost.**
>
>     Thanks for pointing out this problem. We have fixed it in the revised version (Lines 135-161) by highlighting and summarizing each paragraph, such as “Background on URIAL”, “Experimental setup”, and “Results”.
> - **The abbreviation URAIL is not defined.**
>
>     We have added to the paper (Line 058-060) that URIAL stands for “Untuned LLMs with Restyled In-context ALignment”.
> - **Line 139: If providing multi-turn examples improves performance at multi-turn conversation, why do the authors not study it in this paper?**
>
>     We decided to initially focus on single-turn conversations since 1) these are the ones used by URIAL, and 2) already in this setup, as we have shown in our experiments, there are many factors (decoding parameters, number of examples, datasets, etc.) that influence the performance of alignment via ICL, which deserved to be studied carefully.
>
>     Following your request, we have now added an exploration of multi-turn alignment via ICL for two open source LLMs, Mistral-7B-v0.2 and Llama-3.1-8B, and ablated factors like the number of multi-turn conversations and the format of in-context demonstrations. We presented our findings in Section 2.4 (Line 267-308) in our revised version. The results in Table 3 suggest that the presence of multi-turn examples in the context always improves the model’s capability of doing multi-turn conversations as measured by 2nd-turn MT-Bench score. A context with good formatting (i.e., which clearly separates single-turn and multi-turn examples into two groups) leads to better second-turn performance, improving from 5.88 to 6.31 on Mistral-7B-v0.2 and from 6.43 to 6.71 on Llama-3.1-8B. However, further increasing the number of multi-turn examples in the context does not result in better performance.
>
>     We think that the addition of this evaluation complements our previous results, and provides an even more comprehensive understanding of the ICL for alignment.
> - **The results in Figure 2 are not representative in my opinion since the confidence bounds are overlapped largely.**
>
>     Note that we only conclude from Figure 2 that there is an increasing trend in performance with respect to the number of examples. We see this result as motivation for exploring many-shot ICL. Regarding the overlapping confidence bounds, we fully agree as the comment in the caption suggests (i.e., *"The set of rules does not seem to influence the results"*). Thus, we believe that our interpretation of Figure 2 is correct.
> - **Line 320: You say including examples from SkillMix is better than including examples from URIAL. Does URIAL come with a curated dataset as well?**
>
>     The examples from the URIAL prompt are manually created and do not come from any curated datasets. We emphasized here that there is nothing particularly special in the URIAL in-context examples (except their quality and style), and even better instruction-following performance can be achieved by sampling from high-quality datasets like SkillMix.
> - **Figure 3: I don't get it clearly. Is "Llama-3.1-8B-instruct" a model with instruction fine-tuning? Are these rest of ICL methods based on Llama-3.1-8B base models?**
>
>     Yes, Llama-3.1-8B-Instruct is the model released by Meta that was obtained by doing **instruction fine-tuning** and other alignment methods as described in their technical report. Conversely, the ICL methods use the Llama-3.1-8B **base** model.
>
> We thank again the reviewer for the feedback. We hope our rebuttal addresses the raised concerns.

---

> ### Author Response · Authors · 2024-11-29
> **Have our responses addressed your concerns?**
>
> Dear Reviewer ZLSn,
>
> Thank you again for your review and comments. Since the discussion period is closing soon, we are wondering whether our responses have addressed your concerns. If not, we would be happy to continue the discussion and provide further clarifications or explanations.

---

> ### Comment · Reviewer_ZLSn · 2024-11-29
>
> Thanks for addressing my questions. I've increased my score.

---

### Author Response · Authors · 2024-11-25
**General Response and Manuscript Updates**

We thank all the reviewers for the thoughtful feedback, which has helped to improve our paper. Below, we outline the key updates to our manuscript, which address the following:
- **Multi-turn alignment via ICL**: We investigated the impact of multi-turn in-context examples on general instruction following performance of in-context aligned LLMs, taking into account formatting and the quantity of examples. We add Section 2.4 and Table 3 to present the results and our analysis.
- **Revisit our main findings on instruct models**: We reproduced three primary experiments, originally performed on base LLMs, with instruct models: (a) influence of the individual components of URIAL on instruct models (Appendix B.1, Table 7), (b) importance of question-answer matching of demonstrations for in-context alignment on instruct models (Appendix B.2, Table 8), and (c) many-shot in-context alignment on instruct models (Appendix B.3, Figure 10). We find that adding examples in the context consistently declines the instruction-following performance of instruct models, even when using correct URIAL prompt and increasing the number of high-quality examples.
- **KL-divergence between the base LLMs and its aligned counterparts via ICL or IFT**: We carefully examined the difference on probability distribution of next-token prediction between the base LLMs and aligned models by calculating KL-divergence based on models’ output logits. We summarize our results and present analysis in Appendix C and Figure 11.
- **Manuscript clarifications**: extended descriptions about the URIAL prompt (Section 2.1) and the SkillMix dataset (Section 3), summarized our main contributions at the end of Introduction, added a few examples of URIAL prompt and SkillMix data for illustration purposes (Figure 6, 7, 9 in Appendix).

---

### Meta-Review · Area_Chair_8Lwn · 2024-12-18

**Metareview:**

This paper performs a comprehensive study of in-context alignment based on the method of URIAL. The empirical experiments find that in-context alignment generally underperforms compared to instruction fine-tuning. However, the paper then reported that using high-quality carefully selected in-context demonstrations can improve the alignment performance to reduce the gap with instruction fine-tuning.

The reviewers generally agree that the experiments are comprehensive and the conclusions from the experiments are practically useful and can inspire future works in this topic.

Multiple reviewers pointed out the lack of evaluations for multi-turn conversations in the original paper. The authors have added such experiments during rebuttal, and the results look promising (e.g., adding multi-turn examples indeed improve the performance). An important concern pointed out by Reviewer GajX is that the MT-Bench dataset, which is used in the majority of the experimetns in the paper, has only 80 data points, which may call into question the reliability of the findings. However, the authors' response look reasonable to me: the MT-Bench dataset is a widely used and reliable dataset; the authors used AlpacaEval 2.0 to further validate their findings and added more such experiments after rebuttal; the budget concern also looks reasonable to me.

Overall, I think the findings in this paper are insightful and useful for potential future works in this area. Therefore, acceptance is recommended.

**Additional Comments On Reviewer Discussion:**

The most important topic of discussion during the rebuttal phase was whether the MT-Bench dataset is good enough to make the findings in the paper reliable. As explained above, I think the relatively small number of data points in this dataset does not significantly diminish the significance of the findings in this paper, which can provide inspirations for future works along this line.

---

### Decision · Program_Chairs · 2025-01-22

Accept (Poster)